# The Sailor diagram. A new diagram for the verification of two-dimensional vector data from multiple models.

Jon Sáenz[1, 2], Sheila Carreno-Madinabeitia[3], Ganix Esnaola[4, 2], Santos J. González-Rojí[5,6], Gabriel Ibarra-Berastegi[7, 2], and Alain Ulazia[8]

[1]Dept. Applied Physics II, Universidad del País Vasco/Euskal Herriko Unibertsitatea (UPV/EHU), Barrio Sarriena s./n., 48940-Leioa, Spain
[2]Joint Research Unit BEGIK, Instituto Español de Oceanografía (IEO)-Universidad del País Vasco/Euskal Herriko Unibertsitatea (UPV/EHU), Plentziako Itsas Estazioa, Areatza Pasealekua, 48620-Plentzia, Spain
[3]TECNALIA, Basque Research and Technology Alliance (BRTA), Parque Tecnológico de Álava, Albert Einstein 28, E-01510 Vitoria-Gasteiz, Spain
[4]Nuclear Engineering and Fluid Mechanics Dept., Gipuzkoako Ingeniaritza Eskola, Europa Plaza 1, 20018-Donostia, Spain
[5]Oeschger Centre for Climate Change Research, University of Bern, 3010 Bern, Switzerland
[6]Climate and Environmental Physics, University of Bern, 3010 Bern, Switzerland
[7]Nuclear Engineering and Fluid Mechanics Dept., Escuela de Ingeniería de Bilbao, Plaza Ingeniero Torres Quevedo 1, 48013-Bilbao, Spain
[8]Nuclear Engineering and Fluid Mechanics Dept., Gipuzkoako Ingeniaritza Eskola, Otaola etorbidea 29, 20600-Eibar, Spain

**Correspondence:** Jon Sáenz (jon.saenz@ehu.eus)

**Abstract.** A new diagram is proposed for the verification of vector quantities generated by multiple models against a set of observations. It has been designed with the objective, as in the Taylor diagram, of providing a visual diagnostic tool which allows an easy comparison of simulations by multiple models against a reference dataset. However, the Sailor diagram extends this ability to two-dimensional quantities such as currents, wind, horizontal fluxes of water vapour or other geophysical variables by adding features which allow to evaluate directional properties of the data as well. The diagram is based on the analysis of the two-dimensional structure of the mean squared error matrix between model and observations. This matrix is separated in a part corresponding to the bias and the relative rotation of the two orthogonal directions (empirical orthogonal functions, EOFs) which best describe the vector data. Since there is no truncation of the retained EOFs, these orthogonal directions explain the total variability of the original dataset. We test the performance of this new diagram to identify the differences amongst the reference dataset and a series of model outputs by using some synthetic datasets and real-world examples with time-series of variables such as wind, current and vertically integrated moisture transport. An alternative setup for spatially varying time-fixed fields is shown in the last examples, where the spatial average of surface wind in the Northern and Southern Hemispheres according to different reanalyses and realizations from ensembles of CMIP5 models are compared. The Sailor diagrams presented here show that it is a tool which helps in identifying errors due to the bias or the orientation of the simulated vector time series or fields. The R implementation of the diagram presented together with this paper allows also to easily retrieve the individual diagnostics of the different components of the mean squared error and additional diagnostics which can be presented in tabular form.

## 1   Introduction

It has been a long time since visual tools were recognized as an easy way to analyse different properties of datasets. This appreciation is at the root of simple and effective visualizations for exploratory data analysis such as the well-known Hovmöller diagram (Hovmöller, 1949) and the Box Plot (McGill et al., 1978). A visual tool for presenting temperature anomalies has also been recently recognized as a very effective way of presenting information regarding the evolution of climate to general audiences (Hawkins et al., 2019). Visual tools are very helpful in the scientific inquiry, see, for instance Peircean diagrammatic
thinking (Dörfler, 2005). Furthermore, the visualization via diagrammatic representations does not constitute only a way of interpretation. Peircean theory of signs and other studies on scientific creative thinking show that diagrams, together with analogy or extreme thinking, also constitute a way of reasoning and knowledge generation (Dörfler, 2005; Ulazia, 2016).

Visual representation of data allows a fast and intuitive interpretation of many of their characteristics. This has led to the development of many special types of diagrams, particularly in the field of model verification. These diagrams present different
measures of forecast quality as in the case of the well known Relative Operating Characteristic curve (Wilks, 2006) or a combination of Success Ratio and Probability of Detection (Roebber, 2009) to name a few.

Boer and Lambert (2001) designed a diagram based on second-order space-time differences between model simulations and observations as a tool to diagnose the performance of climate models. Their diagram was based on simple quantities such as mean square differences, variances and Pearson's correlation coefficient between observations and model runs. They used the
analytical relationship between the standard deviation of the datasets, their common correlation coefficient and the squared difference between the datasets. They also showed that the diagram could be used for the evaluation of model ensembles.

Following a similar reasoning, Taylor (2001) presented a diagram which has become a well known and popular tool for the evaluation of model simulations against observed data (in general, a *reference* dataset). In the so-called Taylor diagram, the horizontal axis represents the standard deviation of the reference dataset, the radial distance represents the standard deviation
ratio of the forecast against the reference and the angular distance from the $X$ axis is related to the correlation coefficient between the reference dataset (also referred to as observations) and every model run. The distance from the point assigned to a model in the diagram to the point representing the reference dataset is related to the centered root mean squared error. In the Taylor diagram, every model tested is represented by a point in the diagram and visual inspection allows to easily determine which points are closer (i.e. present lower error) to observations. This approach works for any number of models and, therefore,
comparing models using the Taylor diagram is in general faster and easier than using an equivalent table listing the different error measures. This explains the success of the diagram, as shown by the fact that the paper describing it has been cited more than 2300 times at the time of writing this contribution. This diagram is a tool that helps in the fast diagnose of the relative merits of the models. Aspects such as under or overestimated variance, incorrect phasing of the seasonal cycle and many others are reflected in the relative position of the points characterizing a model in the diagram. The diagram is flexible enough so
that it can be extended to ensembles of models. More specific developments such as incorporating bootstrap techniques for the

estimation of confidence intervals can be easily done (González-Rojí et al., 2018; Ulazia et al., 2017) and stress the idea of flexibility associated to the Taylor diagram. Finally, since observed data also suffer from errors, an estimation of the relevance of these observational errors in different datasets can also be achieved by cheking alternative measured datasets against the same reference as if they were models too (Fernández et al., 2007). Thus, the dispersion amongst observational datasets yields an estimate of the uncertainty of the observations (González-Rojí et al., 2019).

Pearson's correlation coefficient between two scalars plays a fundamental role in the design of Taylor's diagram, but there does not exist a single universally accepted definition of the correlation coefficient in two dimensions. Jupp and Mardia (1980) recognized that any multivariate definition of a correlation coefficient equivalent to Pearson's one must be invariant to rotation, be close to zero for independent datasets, smaller or equal than a constant and equal to that constant only if the datasets are related to each other by means of a function. Since they based their definition on these properties, they found that the sum of the squared canonical correlations was a potential definition of the squared correlation coefficient that met the previous requisites. In a previous paper, Cramer (1974) defined the two-dimensional correlation coefficient by means of the product of the canonical correlations. In this case, a low canonical correlation yields a low correlation coefficient because of the use of the product.

Stephens (1979) defined two versions of correlation between vectors by means of functions which satisfy the requirement that two perfectly correlated vector sets can be related by means of an orthogonal transformation. In this case, the vectors are assumed to share a common center and to be unit vectors, so that this measure cannot be used to identify biases between datasets or different standard deviations. In any case, the author correctly asserted that invariance to rotation does not lead to a unique definition of correlation coefficient for multivariate datasets.

Robert et al. (1985) presented an interesting review of different alternatives to compute the correlation coefficient for vector quantities. They recognized that two approaches to the problem exist. The first one is based on the use of canonical correlations between multivariate datasets. In the second approach, the definition of a two-dimensional correlation coefficient for vector datasets is based on functions which satisfy some desirable properties, such as the invariance of the correlation to the rotation of the original datasets or the existence of a limit constant for linearly related vectors as earlier suggested by Jupp and Mardia (1980).

Despite these many previous studies, it is a fact that up to day, several alternative versions of correlation coefficients between vectors exist. The fact that the definition of a two-dimensional correlation coefficient must satisfy the properties mentioned before was also followed by Crosby et al. (1993), who presented an in-depth review of previous definitions in oceanography and meteorology such as Kundu (1976). Crosby et al. (1993) also stated different possible definitions of the correlation coefficient. Amongst them, they proposed a definition similar to the one used by Jupp and Mardia (1980). This definition was later applied to real marine and atmospheric data sets by Breaker et al. (1994) and Cosoli et al. (2008), for instance. A similar result is obtained for the case of complex correlation coefficients (Schreier, 2008). In this case, too, literature (Hanson et al., 1992; Schreier, 2008) shows that there is not a unique definition of the complex correlation coefficient. One of the potential definitions is the one based in canonical correlation analysis, connected to the minimum squared error and highest mutual information

in the signals being compared. This result is consistent with the definition of $R^2$ by Jupp and Mardia (1980) or Crosby et al. (1993).

However, the diagram designed by Taylor (2001) for scalar variables, is being used by modellers when comparing vectorial quantities of model output with observations. For example, Lee et al. (2013) presented a comparison of CMIP3, CMIP5, reanalysis and satellite-based estimations of wind stress by means of the average of the Taylor diagrams for the zonal and

meridional components of the wind stress as a way to apply Taylor diagrams for vector quantities. A different strategy is followed, for instance, in Jiménez et al. (2010). In this case, the behaviour of several models for the zonal and meridional components is not the same in terms of the identification of the model rankings. The best model for the zonal component in terms of its Taylor diagram is not the best one for the meridional component (see their Figure 6). This is a typical problem which arises when using the Taylor diagram with vector data, as also shown in a study about currents measured by means

of an HF-Radar (Lorente et al., 2015). It also appears in the evaluation of global climate models using zonal and meridional components of wind speed (Martin et al., 2011) or in an analysis of moisture fluxes (Ibarra-Berastegi et al., 2011). A last example appears when wind stress components are analyzed (Chaudhuri et al., 2013). A different alternative which allows the use of the Taylor diagram for verification of wind estimations against observations is to use it as a tool to verify the magnitude of the wind (Ulazia et al., 2016, 2017; Rabanal et al., 2019). However, even in this case, the results are limited, since the

information regarding errors in the direction of the vectors is lost.

In a recent paper, Xu et al. (2016) proposed a new method to overcome the defficiencies of the Taylor diagram for vector datasets and produced a new type of diagram visually equal to the original Taylor diagram, but which can be used for vector quantities. It is constructed on the basis of pattern similarities of vector observations and model runs and they call it Vector Field Evaluation (VFE) diagram. It is constructed from both components of the vectors which appear in the vector datasets that are

used for the verification. In order to arrive to the same structure of the Taylor diagram, the authors apply some normalizations to the original two-dimensional vector quantities.

However, in the original paper by Crosby et al. (1993), the authors show that two-dimensional fields showing a perfect correlation according to their definition do not have to be simple two-dimensional counterparts of what we expect in the one-dimensional case (see their Figure 3). Thus, instead of trying to simply replicate the structure of the original Taylor diagram, we

have decided to follow a new approximation which gives more information about the structure of the two-dimensional errors between vector quantities derived from models and their observational counterpart (reference dataset). In order to achieve this goal, we have based our definition in the analysis of the two-dimensional structure of the mean squared error (MSE) between both vectorial datasets. This does not allow us to reduce our diagram to the well-known Taylor diagram used for scalars, as the one produced by Xu et al. (2016). However, we hope that our diagram will be considered a valuable contribution to the set of

techniques used for the evaluation of models, as it visually explores other properties of the error between the vector datasets, such as the relative rotation of the major axes of variability and the underestimation (or overestimation) of variance along each principal axis of the covariance matrix. As will be shown in this contribution, this is an important diagnostic error which would otherwise be lost.

Empirical orthogonal functions are commonly used in the literature for the decomposition of geophysical fields in their temporal and spatial variability (Hannachi et al., 2007). The use of an EOF-based decomposition of a geophysical field is particularly relevant because it produces linear combinations of the original variables (principal components) which are uncorrelated, thus leading to better basis for subsequent stages of the analysis. These uncorrelated principal components are important bricks in the development of statistical analyses based in canonical correlation or multiple regression models, for instance (Barnett and Preisendorfer, 1987; Bretherton et al., 1992). Besides that, these linear combinations are also able to explain decreasing fractions of variance, so that the EOFs form an interesting orthogonal basis for data compression and dimensionality reduction (Monahan et al., 2009). However, the reduction in variance is achieved by truncating the amount of EOFs that are kept for the analysis to a number of EOFs lower than the rank of the corresponding covariance matrix. In the case of our paper, as will be discussed later, the original $2 \times 2$ covariance matrix is expressed by two EOFs, so that there is no truncation in the process, as discussed in detail in Subsection 3.1.

It is the authors' need to find a solution to problems found in the past when using the Taylor diagram for vector quantities that inspired this proposal. The Sailor diagram provides a full analysis of the two-dimensional covariance matrix of the observed and simulated vector fields and, at the same time, it yields exact numerical estimations of the RMSE between those vector fields. Additional diagnostics presented in this contribution such as the relative rotation of the principal axes can be obtained following our methodology. Thus, this contribution provides a useful tool for the verification of simulated vector fields.

We propose the name Sailor diagram as a joke due to the fact that it is a diagram which can be used for winds and currents (properties of geophysical fluid dynamics that sailors need to know about) and because this name is very similar to the original Taylor diagram. Thus, the name can be derived from the original Taylor just by changing two letters in the word (two letters equal the number of dimensions used in the diagram) following the idea behind Lewis Carroll's word ladder puzzles.

Section 2 presents the datasets that we have used as examples of application of our Sailor diagram. Section 3 explains the methodology that we follow to build the two-dimensional diagram. Results are included in Section 4, followed by some concluding remarks in Section 5.

## 2 Data

In order to show that the diagram that we propose is of general interest and can be applied in different studies involving vector magnitudes, we have selected some examples ranging from evident variables (wind or ocean currents) to additional postprocessed quantities such as vertically integrated moisture transports.

### 2.1 Wind data

The first wind dataset that will be used in this paper corresponds to a one-year long dataset of hourly wind (zonal and meridional components) from ERA5 reanalysis at the point $38°$ N, $-124°$ W, in front of Los Angeles and we will refer to it as reference (Ref) onwards. In order to produce synthetic models which are affected by individual sources of error, we have prepared a perturbed version of this dataset which we refer to as MOD1 in which we have just added a constant bias of $(4.8, -6.8)$ m

s$^{-1}$. In order to address a second source of error, a change in the simulated direction, we have applied a counterclockwise rotation of $30°$ to the original dataset in order to produce MOD2. The rotation produces a change in the principal axes of the distribution of zonal and meridional wind and a new bias too, since it rotates the original averaged wind. A third source of error (lack of temporal correlation) is addressed by resampling (without repetition) the original Ref dataset to produce MOD3, which is characterized by perfect mean wind (no bias) and direction of major and minor axes of the distribution of wind but no correlation of wind events. A final synthetic dataset (MOD4) is produced by scaling the wind distribution with a constant factor (2) so that both the mean and the standard deviations of wind are affected.

Next, offshore wind data are also used as our first example of a Sailor diagram constructed with realistic data. The wind dataset (zonal and meridional components) extends from 01/01/2009 to 01/01/2015 and includes five sources (Ulazia et al., 2017). Two Weather Research and Forecasting Model (WRF) simulations around the Iberian Peninsula are used, one with 3DVAR data assimilation every six hours (experiment D) and the second one without data assimilation (experiment N). ERA Interim (ERAI) data (Dee et al., 2011) were also used to nest the two (N and D) WRF runs and these data are also compared with observations. Fully assimilated Level 3 wind analysis data from the second version of Cross-Calibrated Multi-Platform (CCMPv2) are also used (Hoffman et al., 2003; Atlas et al., 2011) for the evaluation. The previous sources will be validated against in-situ observations provided by the buoy in Dragonera, near the Balearic Islands, a buoy managed by Spanish State Ports Authority (Puertos del Estado) (P.P.E., 2015).

## 2.2 Ocean currents

Three different data sources of ocean surface horizontal vectorial currents are also compared with in-situ data. They cover the Bay of Biscay area and include in-situ observations from a deep-water buoy, remotely sensed surface HF-Radar currents and an ocean modelling product. Observational products, both in-situ buoy (named DONOSTIA buoy) and remotely sensed radar currents, belong to the Basque Meteorological Agency (EUSKALMET) and were obtained from https://www.euskoos.eus. They provide hourly data that is punctual in the case of the buoy (approx. location $43.6°$ N and $2.0°$ W). In the case of the HF Radar dataset, it consists of a gridded dataset which covers the corner of the Bay of Biscay (approx. location 43.5-44.7$°$ N and 3.2-1.3$°$ W) with 5 km spatial resolution (Rubio et al., 2011, 2013; Solabarrieta et al., 2014). The ocean modelling product used in this example is the global analysis and forecast product of the Copernicus Marine Environment Monitoring Service (CMEMS), available through their data portal (identifier `GLOBAL_ANALYSIS_FORECAST_PHY_001_024`) (Madec and the NEMO team, 2008; Lellouche et al., 2018).

## 2.3 Vertically integrated water vapour transports

Zonal and meridional components of vertically integrated water vapour transport have been calculated or downloaded from different sources. First, observations were obtained from the sounding data for A Coruña (Station ID 08001, longitude $-8.41°$ E and latitude $43.36°$ N) with a temporal resolution of 12 hours for the period 2010-2014. Both components of vertically integrated moisture transport from ERAI in the original vertical levels of the ECMWF model were downloaded by means of the Meteorological Archival and Retrieval System (MARS) repository at ECMWF at the nearest point to A Coruña.

Both moisture transport components were also calculated using the moisture and wind data from the previously mentioned N and D simulations created with the WRF model over the Iberian Peninsula as described by González-Rojí et al. (2018). The components of the moisture transport were calculated at the nearest point in WRF's grid by means of the vertical integration of the specific humidity (Sáenz et al., 2019) and the zonal and meridional winds over the original 51 $\eta$ levels of the WRF model.

## 2.4 Verification of spatial vector fields

An important application of the Taylor diagram is the verification of climate models and, as such, it is often used to verify the spatial structure of climate model outputs. In order to show that the Sailor diagram proposed in this paper can also be applied for this purpose, some reanalyses are compared. The original NCEP/NCAR first generation reanalysis (Kalnay et al., 1996) is compared to more modern reanalyses such as MERRA2 (Gelaro et al., 2017), CFSv2 (Saha et al., 2014), ERAI and ERA5 (Hersbach et al., 2018). In all those cases, we have analyzed the January average of the monthly values covering a common period (2011-2018), regridded by means of bilinear interpolation to the grid corresponding to the NCEP/NCAR reanalysis case ($2.5° \times 2.5°$).

Finally, in terms of the application of the diagram to a typical case in the analysis of climate models, we use time-averaged wind speed over the Southern Hemisphere (1979-2005). This case example uses the time average of surface wind obtained from ERA5 as the reference dataset. In order to check the behaviour of the diagram when analyzing ensembles of multimodels, we have also downloaded surface wind fields of the historical forcing experiment contributed by three models from the CMIP5 repository for the same period. The first set includes six realizations by IPSL model, developed at the Institute Pierre-Simon Laplace (Dufresne et al., 2013). The second one (including five realizations) derives from the MIROC model (Watanabe et al., 2010). The third case includes four realizations integrated using the HadGEM2-ES model (Collins et al., 2011) from the Hadley Center. All the models and ERA5 reanalysis gridded fields have been bilinearly interpolated to a common $1.25° \times 1°$ regular longitude-latitude grid. This example is selected to illustrate the way the diagram can be applied for the analysis of ensemble data even if the number of realizations for each model is different.

## 3 Methodology

In this section, we present the derivation of the $2 \times 2$ squared-error matrix that is on the basis of the definition of the diagram that is proposed later. The two dimensional squared error matrix is decomposed in the Empirical Orthogonal Functions (EOFs) corresponding to the covariance matrix defined by the zonal and meridional components of observations (and similarly for the covariance matrix defined by each model). Subsection 3.1 describes the decomposition of the matrix $\mathbf{U}$ corresponding to the reference dataset (observations) in its EOFs. A similar notation will be used later for the decomposition of the matrix $\mathbf{V}$ corresponding to the zonal and meridional components of every model which is being compared to observations. Later, the expansion of the $\mathbf{V}$ matrix corresponding to the model is expressed as a rotation from the the EOFs derived from observations (Subsection 3.2).

## 3.1 Decomposition of U in its EOFs

We consider a time series or spatial field of a two-dimensional vectorial variable such as horizontal wind, vertically integrated moisture transport or horizontal currents, for instance. It has been measured at an observatory or buoy (time series) or it is a time-average over a grid (the case of the evaluation of a climatology derived from climate models). By now, we will consider that we are evaluating a time-series of $N$ samples, but later we will present results where the $N$ represents the number of grid points where a time-averaged field is defined. Note that in the following presentation, $\mathbf{U}$ includes the zonal and meridional components of observations and so does $\mathbf{V}$ for a simulated dataset. The observational dataset is formed by the two-dimensional (zonal and meridional) components of vector measurements $\mathbf{u}_i$, with $i = 1 \ldots N$ arranged as rows in an $N \times 2$ matrix $\mathbf{U}$. The average $\bar{\mathbf{u}}$ of the $\mathbf{u}_i$ time series will be repeated as constant rows in an $N \times 2$ matrix $\bar{\mathbf{U}}$. The $2 \times 2$ covariance matrix from the zonal and meridional components of velocity anomalies in the observations is given by

$$\mathbf{S}_u = \frac{1}{N} \left(\mathbf{U} - \bar{\mathbf{U}}\right)^T \left(\mathbf{U} - \bar{\mathbf{U}}\right) = \begin{pmatrix} S_{xx} & S_{xy} \\ S_{xy} & S_{yy} \end{pmatrix}. \tag{1}$$

According to the traditional use of the EOF decomposition of geophysical fields, the eigenvalues and eigenvectors of the covariance matrix from observations $\mathbf{U}$ can be computed by means of the expression

$$\mathbf{S}_u \mathbf{e}_{ui} = \lambda_{ui} \mathbf{e}_{ui}, \tag{2}$$

with $\mathbf{S}_u$ the covariance matrix in Eq. (1), $\mathbf{e}_{ui}$ the $i$–eth eigenvector of the observational vector field and $\lambda_{ui}$ the corresponding $i$–eth eigenvalue, so that

$$f_{ui} = \frac{\lambda_{ui}}{\sum_{i=1}^{r} \lambda_{ui}} \tag{3}$$

represents the fraction of variance in observations explained by the linear combination of the original variables defined by the $i$–eth eigenvector of the covariance matrix (Monahan et al., 2009). In the general case of the EOF analysis in climatological analyses, the rank of the covariance matrix $r$ in Eq. (3) extends to (at most) the minimum between the number of grid points ($N_g$) and the number of samples in the dataset ($N$). In the general case, in order to achieve a truncation of the original dataset, a number $t$ of EOFs lower than the rank of the covariance matrix ($t < r$) is selected, so that the signal in the subspace that can not be spanned by eigenvectors $\mathbf{e}_{uj}$ with $j = t+1 \ldots r$ becomes the part of the original dataset which is truncated. However, in our use of EOFs below, the original covariance matrix as defined in Eq. (1) is of rank two or full rank for any realistic non-linear flow. Since two EOFs ($t = r = 2$) will be used in the expansion of the datasets, no truncation is applied and the full variance in the original dataset will be analyzed in the equations that follow.

Thus, the $\mathbf{U}$ matrix can be expressed by means of the two empirical orthogonal functions of the original vector data (which constitute a complete basis of the horizontal plane) by using the expression

$$\mathbf{U} = \bar{\mathbf{U}} + \mathbf{P}_u^* \mathbf{\Sigma}_u \mathbf{E}_u^T = \bar{\mathbf{U}} + \mathbf{P}_u \mathbf{E}_u^T, \tag{4}$$

with $\mathbf{P}_u^*$ (an $N \times 2$ matrix) the standardized principal components of the $\mathbf{U}$ data, $\mathbf{\Sigma}_u$ ($2 \times 2$ matrix) the standard deviations ($\sigma_{1u}$ and $\sigma_{2u}$) of the leading and second EOFs of the $\mathbf{U}$ field, $\mathbf{E}_u$ ($2 \times 2$ matrix) the matrix holding the orthogonal rotation-matrix leading to the empirical orthogonal functions of the $\mathbf{U}$ field arranged as columns and $\mathbf{P}_u = \mathbf{P}_u^* \mathbf{\Sigma}_u$ ($N \times 2$ matrix) the variance-holding principal components. Please, note that when the standardized principal components $\mathbf{P}_u^*$ are used, this matrix is always multiplied by the corresponding standard deviations, so that no variance is lost in the process. Thus, the anomalies of wind are computed without any loss of variance as

$$\mathbf{U} - \bar{\mathbf{U}} = \mathbf{P}_u^* \mathbf{\Sigma}_u \mathbf{E}_u^T = \mathbf{P}_u \mathbf{E}_u^T \tag{5}$$

and the corresponding principal components

$$\mathbf{P}_u = \mathbf{P}_u^* \mathbf{\Sigma}_u = \left(\mathbf{U} - \bar{\mathbf{U}}\right) \mathbf{E}_u, \tag{6}$$

and their standardized counterparts

$$\mathbf{P}_u^* = \left(\mathbf{U} - \bar{\mathbf{U}}\right) \mathbf{E}_u \mathbf{\Sigma}_u^{-1}. \tag{7}$$

Unless the wind (current) time series is completely arranged across a straight line (something which is very unlikely in observed vector variables unless the flow is stationary and laminar), $\mathbf{\Sigma}_u$ is a full-rank diagonal matrix:

$$\mathbf{\Sigma}_u = \begin{pmatrix} \sigma_{1u} & 0 \\ 0 & \sigma_{2u} \end{pmatrix}, \tag{8}$$

with $\sigma_{1u} > \sigma_{2u}$. Due to the fact that the rotation matrix is always full rank (in the two-dimensional space spanned by the zonal and meridional components, given enough samples), the $\mathbf{E}_u$ matrix can also be interpreted geometrically as a rotation matrix expressed as a function of the angle $\theta_u$ formed by the leading (second) EOF with the zonal (meridional) axis as:

$$\mathbf{E}_u = \begin{pmatrix} \cos\theta_u & -\sin\theta_u \\ \sin\theta_u & \cos\theta_u \end{pmatrix}, \tag{9}$$

The first column of the $\mathbf{E}_u$ matrix is the first eigenvector of observations in the horizontal plane, $\mathbf{e}_{u1}$. Similarly, the second column of $\mathbf{E}_u$ corresponds to $\mathbf{e}_{u2}$, the second eigenvector of the observational covariance matrix.

The principal components and EOF rotation matrices fulfill the well-known orthogonality properties

$$\mathbf{P}_u \mathbf{P}_u^T = \mathbf{\Sigma}_u^2, \tag{10}$$

so do the standardized principal components

$$\mathbf{P}_u^* \mathbf{P}_u^{*T} = \mathbb{1} \tag{11}$$

and eigenvectors (EOFs) in the horizontal plane

$$\mathbf{E}_u \mathbf{E}_u^T = \mathbf{E}_u^T \mathbf{E}_u = \mathbb{1}. \tag{12}$$

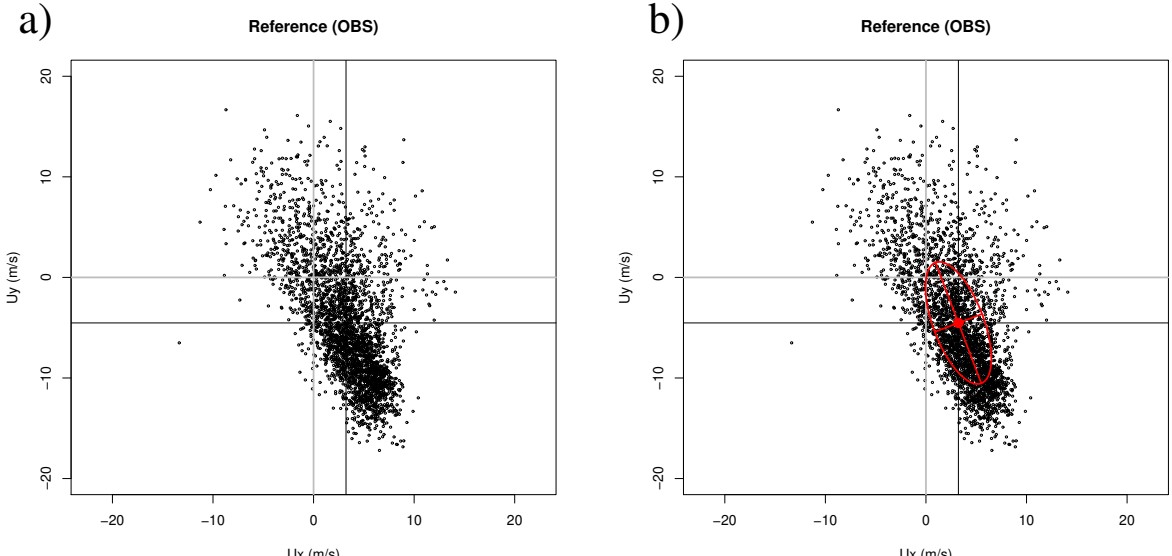

**Figure 1.** Scatterplot of wind in dataset Ref (panel a) and its decomposition in terms of the principal axes corresponding to the covariance matrix of the zonal and meridional components (panel b) as defined in Equation (4).

Figure 1 (panel a) illustrates in a scatterplot the distribution of measurements of zonal and meridional wind components in the Reference dataset and is presented to make easily understandable the next step in the derivation of the Sailor diagram. Panel b in Figure 1 shows on top of the previous scatterplot the ellipses centered in the mean of the reference dataset used in the Sailor diagrams by using the semi-major and semi-minor axes as defined by the EOF decomposition of the two-dimensional covariance matrix of the zonal and meridional components of the original vector field, the directions of the principal axes

(matrix $\mathbf{E}_u$) and the standard deviations corresponding to the principal components $\mathbf{P}_u$. From equations (7) and (11), the quadratic form leading to the ellipses in the diagram can be obtained by applying the Frobenius norm to equation (11) as

$$||\mathbf{P}_u^* \mathbf{P}_u^{*T}||_F = || \left(\mathbf{U} - \bar{\mathbf{U}}\right) \mathbf{E}_u \mathbf{\Sigma}_u^{-2} \mathbf{E}_u^T \left(\mathbf{U} - \bar{\mathbf{U}}\right)^T ||_F = 1. \tag{13}$$

  The principal components are combined according to the quadratic form in Equation (13). This shows that the ellipse produced from the EOF decomposition of the two-dimensional covariance matrix is a good way to make a simple and clear

representation of the original scatterplot. The eccentricity of the ellipse

$$\varepsilon_u = \sqrt{1 - \frac{\sigma_{2u}^2}{\sigma_{1u}^2}} \tag{14}$$

is an interesting indicator for additional diagnostics designed for testing the reliability of rotation angles due to the degeneracy of the eigenvalues.

Following similar notation to the one used for the observations ($\mathbf{U}$ matrix), the time series (or time-averaged constant field over $N$ points in a grid) of simulated wind (current, wave energy flux, vertically integrated moisture transport ...) at the same observatory (or the closest grid point) formed by the two-dimensional (zonal and meridional components) simulations $\mathbf{v}_i$, with $i = 1 \ldots N$ will be arranged as rows in an $N \times 2$ matrix $\mathbf{V}$. The average vector from model data $\bar{\mathbf{v}}$ is arranged as constant rows in an $N \times 2$ matrix $\bar{\mathbf{V}}$. The $\mathbf{V}$ matrix (and its anomalies) can be expressed as done for observations as in Equations (4) and (5) above by means of the empirical orthogonal functions of the two-dimensional covariance matrix from simulated zonal and meridional components of wind (current, moisture transport ...) data

$$\mathbf{V} = \bar{\mathbf{V}} + \mathbf{P}_v^* \mathbf{\Sigma}_v \mathbf{E}_v^T = \bar{\mathbf{V}} + \mathbf{P}_v \mathbf{E}_v^T \quad \Leftrightarrow \quad \mathbf{V} - \bar{\mathbf{V}} = \mathbf{P}_v^* \mathbf{\Sigma}_v \mathbf{E}_v^T = \mathbf{P}_v \mathbf{E}_v^T, \tag{15}$$

with equivalent interpretations and equal ranks for $\mathbf{P}_v^*$, $\mathbf{\Sigma}_v$, $\mathbf{E}_v$ and $\mathbf{P}_v = \mathbf{P}_v^* \mathbf{\Sigma}_v$ as presented before for observations.

## 3.2 Expansion of the matrix V in the EOFs defined by observations

In general, the mean values and EOFs derived from observations ($\mathbf{U}$) and simulations ($\mathbf{V}$) will not be the same. This is shown in Figure 2, with panel a clearly showing a change in the bias between both datasets and a counter-clockwise rotation for the case of panel b, as expected from the way these synthetic datasets were produced. It is clearly seen that in the case of MOD1 the structure of the covariance matrix has not changed, whilst a different orientation (but no scaling of the semi-major and semi-minor axes) appears in the case of MOD2.

In order to identify this kind of errors (derived from rotations of the axes), the orthonormal EOFs in the $\mathbf{E}_v$ matrix can be expressed as the result of a rotation applied to the EOFs derived from the observations (accepting these as *true* EOFs). Thus, the rotation matrix $\mathbf{R}_{vu}$ is defined by an angle $\theta_{vu} = \theta_v - \theta_u$ as

$$\mathbf{R}_{vu} = \begin{pmatrix} \cos\theta_{vu} & -\sin\theta_{vu} \\ \sin\theta_{vu} & \cos\theta_{vu} \end{pmatrix}, \tag{16}$$

so that

$$\mathbf{E}_v = \mathbf{R}_{vu} \mathbf{E}_u, \tag{17}$$

$$\mathbf{V} = \bar{\mathbf{V}} + \mathbf{P}_v \mathbf{E}_u^T \mathbf{R}_{vu}^T \tag{18}$$

and the corresponding principal components can be expanded as

$$\mathbf{P}_v = \left( \mathbf{V} - \bar{\mathbf{V}} \right) \mathbf{R}_{vu} \mathbf{E}_u = \tilde{\mathbf{V}} \mathbf{E}_u, \tag{19}$$

with $\tilde{\mathbf{V}} = \left( \mathbf{V} - \bar{\mathbf{V}} \right) \mathbf{R}_{vu}$ representing the model-based $\mathbf{V}$ anomalies *rotated* to the basis given by the EOFs corresponding to observations.

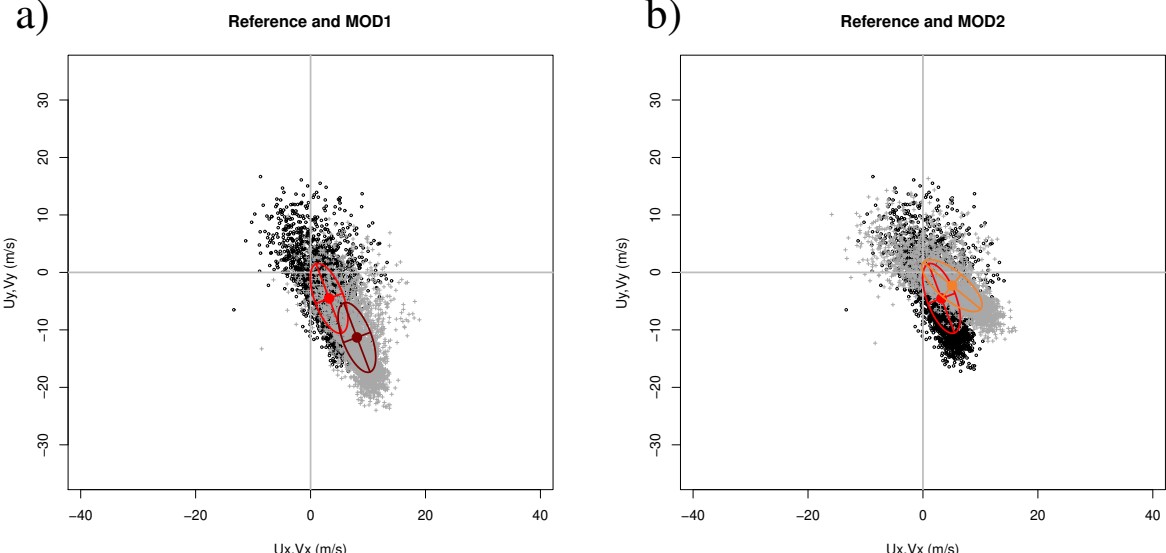

**Figure 2.** Scatterplot of wind in datasets Ref (panel a, black circles) and MOD1 (panel a, grey crosses) and their decomposition in terms of the principal axes corresponding to the covariance matrix of the zonal and meridional components of each dataset as defined in Equations (12) and (15). The comparison of the reference dataset (black circles) with model MOD2 (grey crosses) is shown in panel b.

Since both $\mathbf{e}_{u1}$ and $-\mathbf{e}_{u1}$ are solutions of the eigenvalue equation when the diagonalization of the two-dimensional covariance matrix is performed (the same happens with $\mathbf{e}_{v1}$ and $-\mathbf{e}_{v1}$ for model data), $\theta_{vu}$ may take difficult to understand values even for eigenvectors which span similar subspaces. This is due to the fact that both $\theta_{vu} = 0$ and $\theta_{vu} = \pi$ refer to eigenvectors that point in perfect directions. In order to provide an easier to interpret diagnostic of the adequacy of the EOFs from observations and model, the absolute value of the congruence coefficient (Cheng et al., 1995) can also be used. It is defined as

$$g_{ii} = |\mathbf{e}_{ui} \cdot \mathbf{e}_{vi}| \tag{20}$$

and measures the agreement between the pairs of EOFs from observations ($\mathbf{e}_{ui}$) and models ($\mathbf{e}_{vi}$). Since this coefficient equals the cosine of the angle between both directions, and since the absolute value is used, the closest its value is to one, the best agreement exists between $\mathbf{e}_{ui}$ and $\mathbf{e}_{vi}$. Due to the orthogonality relationship between the EOFs, only the congruence coefficient for EOF1 is computed, since it is equal to the one computed using EOF2 (matrices $\mathbf{E}_u$ and $\mathbf{E}_v$ are orthonormal).

### 3.3 Expansion of the mean-squared error

The $(2 \times 2)$ matrix that represents the mean squared error between the $\mathbf{U}$ and $\mathbf{V}$ datasets is given by

$$\mathbf{\Delta}_{uv}^2 = \frac{1}{N} (\mathbf{V} - \mathbf{U})^T (\mathbf{V} - \mathbf{U}) \tag{21}$$

and the aggregated scalar mean squared error of both components of the vector dataset is given by its Frobenius norm

$$\varepsilon^2 = ||\mathbf{\Delta}_{uv}^2||_F. \tag{22}$$

Substituting Eq. (4) and Eq. (15) into Eq. (21), it can be shown that

$$\mathbf{\Delta}_{uv}^2 = \frac{1}{N}\mathbf{B}_{uv}^2 + \frac{1}{N}\left(\mathbf{S}_{uv}^T + \mathbf{S}_{uv}\right) + \frac{1}{N}\mathbf{D}_{uv} = \frac{1}{N}\mathbf{B}_{uv}^2 + \frac{1}{N}\mathbf{C}_{uv} + \frac{1}{N}\mathbf{D}_{uv} \tag{23}$$

with

$$\mathbf{B}_{uv}^2 = \left(\bar{\mathbf{V}} - \bar{\mathbf{U}}\right)^T \left(\bar{\mathbf{V}} - \bar{\mathbf{U}}\right), \tag{24}$$

$$\mathbf{S}_{uv} = \left(\mathbf{E}_v \mathbf{\Sigma}_v \mathbf{P}_v^{*T} - \mathbf{E}_u \mathbf{\Sigma}_u \mathbf{P}_u^{*T}\right)\left(\bar{\mathbf{V}} - \bar{\mathbf{U}}\right) = \left(\mathbf{E}_v \mathbf{P}_v^T - \mathbf{E}_u \mathbf{P}_u^T\right)\left(\bar{\mathbf{V}} - \bar{\mathbf{U}}\right) \tag{25}$$

and

$$\mathbf{D}_{uv} = \mathbf{E}_u \mathbf{\Sigma}_u^2 \mathbf{E}_u^T + \mathbf{E}_v \mathbf{\Sigma}_v^2 \mathbf{E}_v^T - \left(\mathbf{E}_u \mathbf{\Sigma}_u \mathbf{P}_u^{*T} \mathbf{P}_v^* \mathbf{\Sigma}_v \mathbf{E}_v^T + \mathbf{E}_v \mathbf{\Sigma}_v \mathbf{P}_v^{*T} \mathbf{P}_u^* \mathbf{\Sigma}_u \mathbf{E}_u^T\right) \tag{26}$$

which can also be written using non-standardized $\mathbf{P}_u$ and $\mathbf{P}_v$ principal components as

$$\mathbf{D}_{uv} = \mathbf{E}_u \mathbf{\Sigma}_u^2 \mathbf{E}_u^T + \mathbf{E}_v \mathbf{\Sigma}_v^2 \mathbf{E}_v^T - \left(\mathbf{E}_u \mathbf{P}_u^T \mathbf{P}_v \mathbf{E}_v^T + \mathbf{E}_v \mathbf{P}_v^T \mathbf{P}_u \mathbf{E}_u^T\right). \tag{27}$$

$\mathbf{B}_{uv}^2$ represents the part of the squared error due to the magnitude of the bias vector (difference of both means) between both vector datasets.

The (symmetric) matrix $\mathbf{C}_{uv} = \mathbf{S}_{uv}^T + \mathbf{S}_{uv}$ reflects the error due to the projection of the bias into the differences of vector anomalies. Since the bias matrices are constant, the sum of the projections become the sum of anomalies and, as such, they become zero. This interpretation is clear if Eq. (5) and the corresponding one for the model anomalies are substituted into the definition of the matrix $\mathbf{S}_{uv}$ in Eq. (25), yielding

$$\mathbf{S}_{uv} = \left(\left(\mathbf{V} - \bar{\mathbf{V}}\right) - \left(\mathbf{U} - \bar{\mathbf{U}}\right)\right)^T \left(\bar{\mathbf{V}} - \bar{\mathbf{U}}\right) = \left(\mathbf{V} - \mathbf{U}\right)^T \left(\bar{\mathbf{V}} - \bar{\mathbf{U}}\right) - \left(\bar{\mathbf{V}} - \bar{\mathbf{U}}\right)^T \left(\bar{\mathbf{V}} - \bar{\mathbf{U}}\right) = 0. \tag{28}$$

Since this matrix is zero, $\mathbf{C}_{uv}$ will also be zero.

Finally, the matrix $\mathbf{D}_{uv}$ is related to the covariance matrix of anomalies, as also clearly seen if Eq. (5) and the corresponding one for simulated data are substituted into Eq. (27).

In order to improve the graphical interpretation of the components of the error, the expression of the empirical orthogonal functions of $\mathbf{V}$ as a rotation of the *true* ones (derived from observations $\mathbf{U}$) is used. Thus, considering Eq. (17), the matrix $\mathbf{D}_{uv}$ above can be rewritten in terms of the EOFs corresponding to observations as

$$\mathbf{D}_{uv} = \mathbf{E}_u \mathbf{\Sigma}_u^2 \mathbf{E}_u^T + \mathbf{R}_{vu} \mathbf{E}_u \mathbf{\Sigma}_v^2 \mathbf{E}_u^T \mathbf{R}_{vu}^T - \left(\mathbf{E}_u \mathbf{P}_u^T \mathbf{P}_v \mathbf{E}_u^T \mathbf{R}_{vu}^T + \mathbf{R}_{vu} \mathbf{E}_u \mathbf{P}_v^T \mathbf{P}_u \mathbf{E}_u^T\right). \tag{29}$$

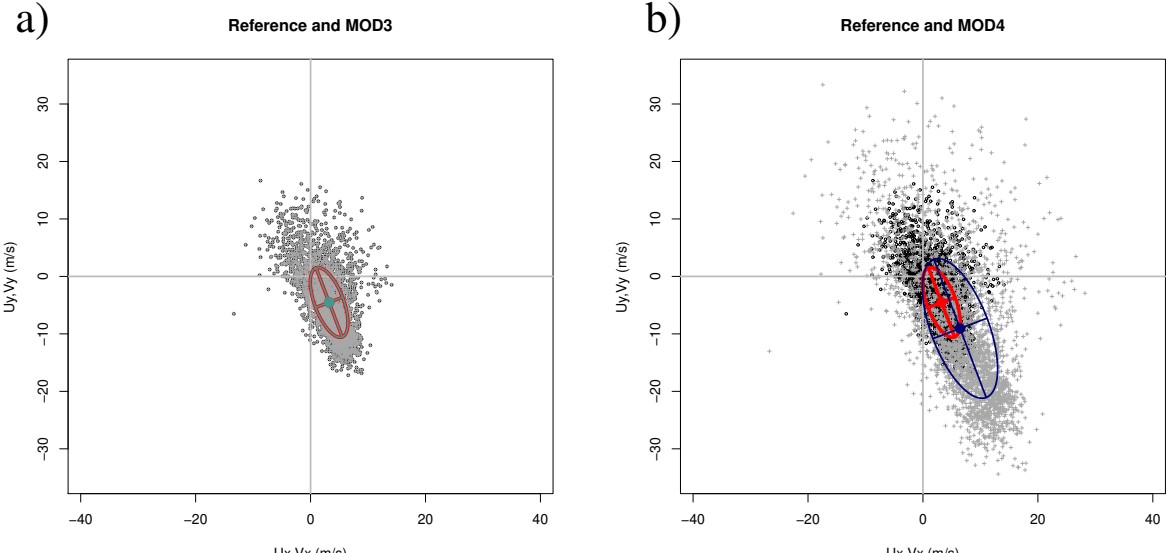

**Figure 3.** Scatterplot of wind in datasets Ref (panel a, black circles) and MOD3 (panel a, grey crosses) and their decomposition in terms of the principal axes corresponding to the covariance matrix of the zonal and meridional components of each dataset. The comparison of the reference dataset (black circles) with model MOD4 (grey crosses) is shown in panel b.

If $\mathbf{\Gamma}_{vu} = \mathbf{P}_u^T \mathbf{P}_v$ is proportional to the covariance between both datasets' principal components, the above expression can be written as:

$$\mathbf{D}_{uv} = \mathbf{E}_u \mathbf{\Sigma}_u^2 \mathbf{E}_u^T + \mathbf{R}_{vu} \mathbf{E}_u \mathbf{\Sigma}_v^2 \mathbf{E}_u^T \mathbf{R}_{vu}^T - \left( \mathbf{E}_u \mathbf{\Gamma}_{vu} \mathbf{E}_u^T \mathbf{R}_{vu}^T + \mathbf{R}_{vu} \mathbf{E}_u \mathbf{\Gamma}_{vu}^T \mathbf{E}_u^T \right). \tag{30}$$

The interpretation of this expression is that all the matrices involved in the mean squared error can be expressed in the axes defined by the leading and second EOFs of the $\mathbf{U}$ (observational) dataset. Thus, using the axes corresponding to the observational
dataset $\mathbf{U}$, we can produce a diagram which gives us a fast visual impression of the structure of the error in two-dimensional variables the same way the Taylor diagram performs for univariate datasets. Therefore, the diagram presented in this contribution includes not only scalar information in the estimation of the error, but also information regarding the main directions of variability of the vectors and their differences by means of the characteristics of the ellipses defined by Eq. (19) from the different datasets.

Figure 3 presents two interesting cases. The first case, MOD3, is implausible from the point of view of a real model, but it constitutes an interesting case study to analyze the properties of the diagram. In MOD3, a simple random permutation of the original observations has been performed. Thus, there are neither bias nor rotations of the principal axes. From the point of view of the graphical example shown, it seems that the model is perfect, but it is not, due to the lack of temporal correlation between model and observations. This is only apparent if the full RMSE is taken into account, as shown by Table 1. Thus, a
legend with the RMSE as defined in Eq. (22) must be added to the plot in order to arrive to precise comparison of datasets. The

comparison of columns $\sigma^2$ and $\sum_i \sigma_i^2$ in Table 1 shows that the full variance of the datasets is taken into account in the EOF decomposition, as both columns present the same values.

| Model | $\sigma^2$ | $\sum_i \sigma_i^2$ | $\theta_u$ | $\theta_v$ | $\theta_{vu}$ | $R^2$ | \|bias\| | RMSE | $\varepsilon$ | $g_{11}$ |
|---|---|---|---|---|---|---|---|---|---|---|
| Ref | 47.56 | 47.56 | 1.93 | | | | | | 0.92 | |
| MOD1 | 47.56 | 47.56 | | 1.93 | 0.00 | 2.00 | 8.34 | 5.56 | 0.92 | 1.00 |
| MOD2 | 47.56 | 47.56 | | 2.46 | 0.52 | 2.00 | 2.88 | 8.69 | 0.92 | 0.87 |
| MOD3 | 47.56 | 47.56 | | -1.21 | 0.72 | 0.00 | 0.00 | 1.52 | 0.92 | 1.00 |
| MOD4 | 190.24 | 190.24 | | 1.93 | 0.00 | 2.00 | 5.56 | 11.76 | 0.92 | 1.00 |

**Table 1.** Individual components of the error for the synthetic datasets used for illustration of the methodology. $\sigma^2$ represents the total variance (m$^2$ s$^{-2}$) of every dataset as computed from the original zonal and meridional components. $\sum_i \sigma_i^2$ represents the variance (m$^2$ s$^{-2}$) of wind for every dataset (reference or model) as computed from the EOF decomposition (axes of the ellipses in the diagrams). $\theta_u$ and $\theta_v$ represent the angles (radians) of the semi-major axes of the ellipses calculated for reference and models. $\theta_{vu}$ (radians) represents the relative rotation of the semi-major axis of the model data with respect to the observations. $R^2$ represents the two-dimensional squared correlation coefficient (sum of the squared canonical correlations). \|bias\| represents the magnitude of the bias (m s$^{-1}$). RMSE holds the root mean squared error (m s$^{-1}$). The eccentricity of the ellipses ($\varepsilon$) is the same for all the synthetic datasets because of the way they have been built. Finally, $g_{11}$ represents the congruence coefficient (Eq. 20) for EOF1 of all models with respect to EOF1 as derived from observations.

On the other side, panel b in Figure 3 shows that for the scaled dataset (MOD4), the sizes of the major and minor axes of the ellipses allow a fast visual detection of the scaling present in the dataset. The individual components of the error for all the synthetic datasets used in the description of the methodology are also presented in Table 1. The eighth column shows the full RMSE between vector fields. It is apparent from this aggregated estimation of error that it properly evaluates the differences due to the lack of correlation that have been mentioned in the case of MOD3 (no bias and perfect orientation and axes of the ellipses) too. The rotation angle (column $\theta_{vu}$ in radians) correctly identifies the way the errors have been introduced in the different synthetic models. Despite the rotation of the ellipses apparent in columns $\theta_u$ and $\theta_v$ (the case of MOD2), the fact that the semiaxes are of the same relative length is clearly seen by the value of the eccentricity $\varepsilon$, which also supports the way the ellipses are presented in Figures 1 to 3. On the other side, the interpretation of the angles is complicated by the fact that both $\mathbf{e}_{ui}$ and $-\mathbf{e}_{ui}$ are a right solution of the eigenvalue problem in Eq. (2). This is apparent in the case of MOD3, in which the eigenvalue problem yields eigenvectors pointing in the same direction with different sign, so that $\theta_v = -1.21 + \pi$ yields the same value as $\theta_u$. The orientation of both eigenvectors is the same for all models except MOD2, as effectively shown by column $g_{11}$ in Table 1, which holds the absolute value of the congruence coefficient.

The different properties of the synthetic datasets presented so far can be abbreviated in Figure 4, which presents in panels a (left) and b (right) uncentered and centered (respectively) versions of the Sailor diagram. In the uncentered version of the Sailor diagram, each ellipse, as defined by Eq. (13), is centered in its own average. This allows an easy interpretation of the bias term. In order to improve the interpretability of the rotation/scaling parameters of the ellipses (semi-major and semi-minor axes and standard deviations), the ellipse corresponding to observations is also drawn in gray centered at the same average of

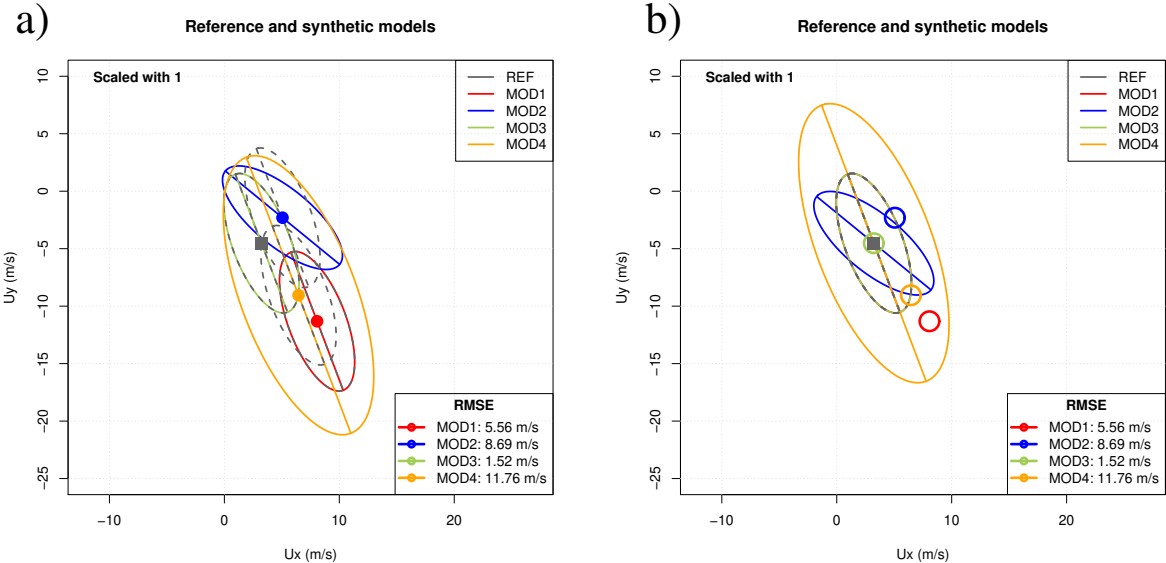

**Figure 4.** Uncentered (left) and centered (right) versions of the Sailor diagram after placing the ellipses from all the synthetic datasets in the same plot.

every model. This way, the rotations and scalings of the vectors produced by models can easily be compared against the ones drawn from observations. However, in some cases (depending on the relative values of the bias and the standard deviations), it might be more interesting to plot all the ellipses centered at the mean corresponding to the observations and identify the bias by means of coloured dots, as shown in the centered version of the diagram (right panel in Figure 4).

An additional reason which supports that the Sailor diagram introduces powerful diagnostics for vector data is properly shown in Table 1. According to the column which shows the squared correlation coefficient, all models show a perfect match ($R^2 = 2$) for the two-dimensional correlation coefficient except the one built by randomly resampling the data (MOD3). However, Figures 2 and 4 clearly show that the wind data in MOD2 is rotated with respect to the reference dataset. This is not detected by $R^2$ because it yields perfect results by construction when there is a linear relationship between both vector

datasets (Crosby et al., 1993). However, an analysis based on the full components of the RMSE as the one performed in the Sailor diagram (Figure 4 and Table 1) clearly highlights these directional problems. The squared two-dimensional correlation $R^2 = 2$ reflects that there is a perfectly linear relationship (rotation in this case). However, a full diagnostic of the differences between observations and model data (such as finding the rotation angle previously mentioned) must involve a full analysis of the directional error.

## 3.4 Extension of the methodology to spatial fields

In the case of the analysis of the ability of models to represent the spatial distribution of an averaged field (a typical use of the Taylor diagram in climatology, for instance), there is no change needed to the diagram defined so far. Instead of using the *T*-mode of principal components (covariance matrix defined by temporal covariances), we can just use the *S*-mode, in traditional terminology of principal components (Compagnucci and Richman, 2008). Thus, in the previous description, $N$ will run along the grid points, and the two-dimensional biases and covariances are computed in the spatial domain, but the error analysis is still being performed onto two-dimensional vectors. As an example of this very common case in the application of Taylor's diagram to climatology, we present an example including the comparison of multi-year averages of Northern Hemisphere surface wind vectors. For the case of spatial grids, an external standard area-weighting by means of factors given by $\sqrt{\cos \phi}$ with $\phi$ latitude (North et al., 1982) is commonly applied to the data in order to avoid an excessive weight in the results of points in polar latitudes which represent much a lower area in a regular longitude-latitude grid.

## 3.5 Use of the diagram with ensembles of models

As a final example, the use of the diagram with a multimodel ensemble is shown. In this case, the long-term (27 years) climatologies of surface wind over the Southern Hemisphere from three models with a different number of realizations are compared with the corresponding climatology from ERA5. As described above, since this also involves a comparison of data on a regular longitude-latitude grid, the covariance matrix is built over the spatial points and the external weights given by $\sqrt{\cos \phi}$ with $\phi$ latitude (North et al., 1982) are applied to avoid an overrepresentation of polar regions in the results.

## 3.6 R package implementing the methodology

The authors have created an R package called `SailoR` which is freely available in the Comprehensive R Archive Network (CRAN). The package has been used to produce the plots presented in section 4, and the code to prepare some of these plots and tables are provided as examples in the manual of the package. Besides producing the diagrams shown as an example in this paper, the package also computes all the individual terms used in the analysis of the MSE error as described in Section 3. Thus, different aspects of the main principal axes, their relative rotation, the two-dimensional correlation coefficient and the combined RMSE can be readily analyzed for different vector datasets and exported to tables which can be presented in publications.

## 4 Use of the elements in the error matrix in the diagram

### 4.1 Wind over a Mediterranean location

The first example of a Sailor diagram built using real data is shown in Figure 5 (left). In it, the $X$ axis represents the zonal component of wind and the $Y$ axis its meridional component. The mean 2D vector corresponding to each of the datasets is represented by a colored circle, except for the reference dataset, which uses a grey square. The leading EOF of the two-dimensional covariance matrix of zonal and meridional components of every dataset is represented by the direction corresponding to the

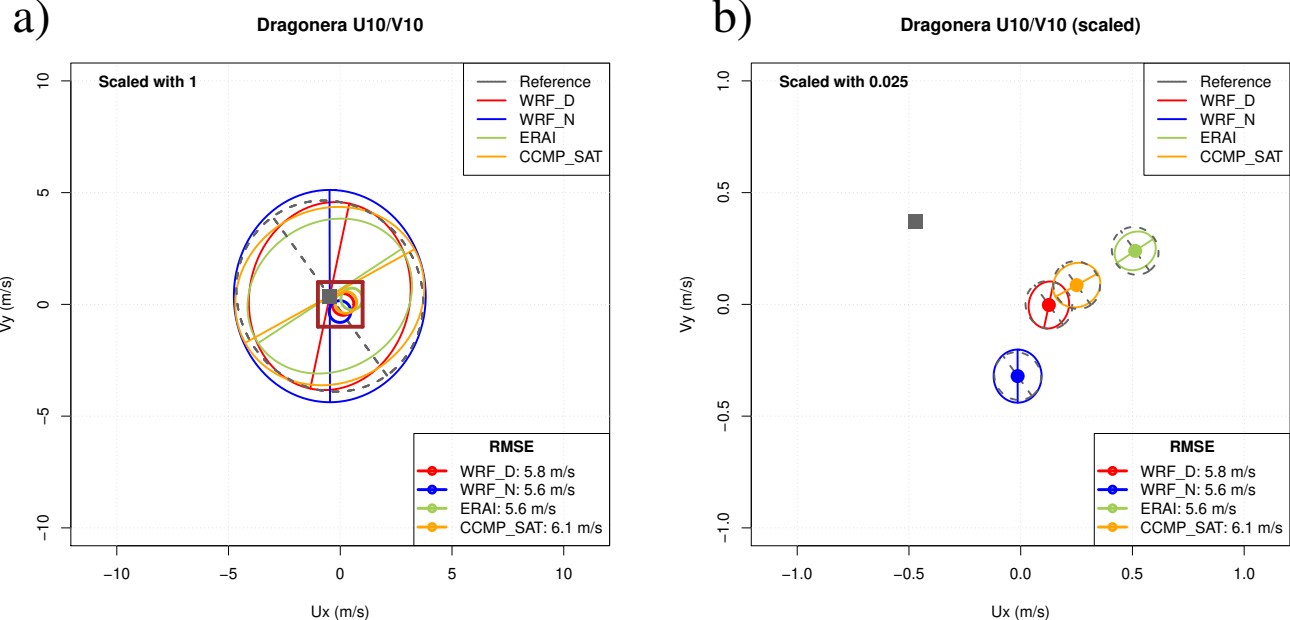

**Figure 5.** Sailor diagram with default parameters (left) and ellipses scaled by a factor 0.025 to improve visibility of the directional error (right) for the wind observed and simulated in Dragonera (buoy in the Mediterranean).

semi-major axis of the ellipse that is plotted centered at every model's mean value (same colour as the one used to represent the model mean). The second EOF to each model is perpendicular to the previous direction by construction due to the orthonormality constraint in Equation (12). The grey ellipse centered at each model mean represents the EOF from the reference dataset (observations). Thus, the angle between the colored and grey semi-major axes represents the relative rotation ($\theta_{vu}$) between EOFs from observations and simulations. The lengths of the semi-major and semi-minor axes (colour and grey) show the vari-

ances explained by each EOF (model and reference) at their principal axes. The comparison of these lengths between coloured and grey ellipses allows to address the question whether the model underestimates or overestimates the variances at each of the principal axes. In this particular example, since the model vs observation biases are much lower than the variance explained by the principal axes defined by the EOFs, the interpretation of this diagram is not very easy. However, it is already showing the main directions of the error matrices, their biases and the position of the reference dataset. The legend at the bottom right

corner shows the total RMSE error given by Eq. (22) in subsection 3, which takes into account both the contribution from the bias (distance of the points to the reference dataset's mean) and the different orientation and lengths of the major and minor axes (EOFs).

     In order to show that different designs optimize the information transmitted by the diagram, in the second diagram prepared using the data from the same example, the ranges of both axes are limited and the ellipses corresponding to the main directions

of the error matrix are accordingly scaled by means of a small scale factor (0.025). The brown square in left panel shows the

area which is amplified in the right panel and it illustrates the role played by the scale factor, which reduces (or amplifies) the size of the axes of the ellipses, thus making easier to appreciate the relative differences in biases while still making possible to get access to the information relative to the rotation of the principal axes. In the scaled version of the diagram (Figure 5, right) it can be seen that the distance between every coloured point corresponding to a given model to the grey square represents the

bias amongst the datasets and they can effectively be visually compared. On the other hand, the grey ellipses and their semiaxes show the main structure of the variability of the reference dataset. This grey ellipse is plotted centered on the point representing the mean of every model, where the EOFs corresponding to that model are also shown for comparison. Both ellipses (the one corresponding to the model being analyzed and the one corresponding to the reference dataset) are scaled by the same scale factor so that they are not deformed during the scaling process. The use of ellipses and their major and minor axes allows to

easily compare the main directions of variability of the observed (grey) and modelled (colored) winds. It shows that the ones corresponding to the WRF model are the closest ones to observations. It can be seen that both WRF simulations show a smaller rotation of their major axes with respect to the one from observations. The model EOFs are almost orthogonal from the ones in observations for the case of ERAI or CCMPv2 (CCMP_SAT in legend). The legend at the bottom right corner, in any case, presents the real RMSE error without scaling its value.

In this particular case, it might seem sensible to think that the fact that the variances of the major and minor axes are very close points to a weakness of the diagram since, in that case, the determination of the angle of the axes will be arbitrary. However, it has to be considered that the final index of agreement would still be the RMSE, which does not depend on the eigenvectors of the covariance matrix. Thus, the results in terms of direction might not be very reliable in case that the eigenvalues are degenerated, but the RMSE is not affected by this problem. Thus, the use of the eccentricity of the ellipses (provided as an

output in our R package) can be useful to diagnose those cases (in which eccentricity is very low) that make estimations of relative rotations difficult. For a more precise determination of the reliability of the rotation angle, a bootstrap analysis of the rotation angles can also be conducted, if needed, since the evaluation of the angles is independent of the production of the diagram.

## 4.2   Surface current in the Bay of Biscay

Figure 6 (left) shows an alternative version of the Sailor diagram. In this particular case, the bias is relatively low. Thus, in order to ease the interpretation of the structure of the errors, the ellipses representing the first and second EOFs are drawn on top of the point corresponding to observations. The fact that the bias is small is only affecting the part of the RMSE derived from term $\mathbf{B}_{uv}^2$ in Eq. (23). As in the previous case, they are scaled (four times larger) in order to improve their visibility. It is clear that the relevant part in terms of the errors of models versus observations is not the bias, but the way the variability is represented,

instead. The HF-Radar data's leading EOF (observational data, actually) is closer to the one from in-situ observations, as could be expected, since both cases represent observational (in-situ versus remote) estimations of currents. In this case, the ellipses clearly show not only the difference in the orientation of the EOFs, but also the underestimation of the variability present both in radar data, but especially in the case of model data. As in the previous case, the legend at the bottom right corner shows unscaled total RMSE errors.

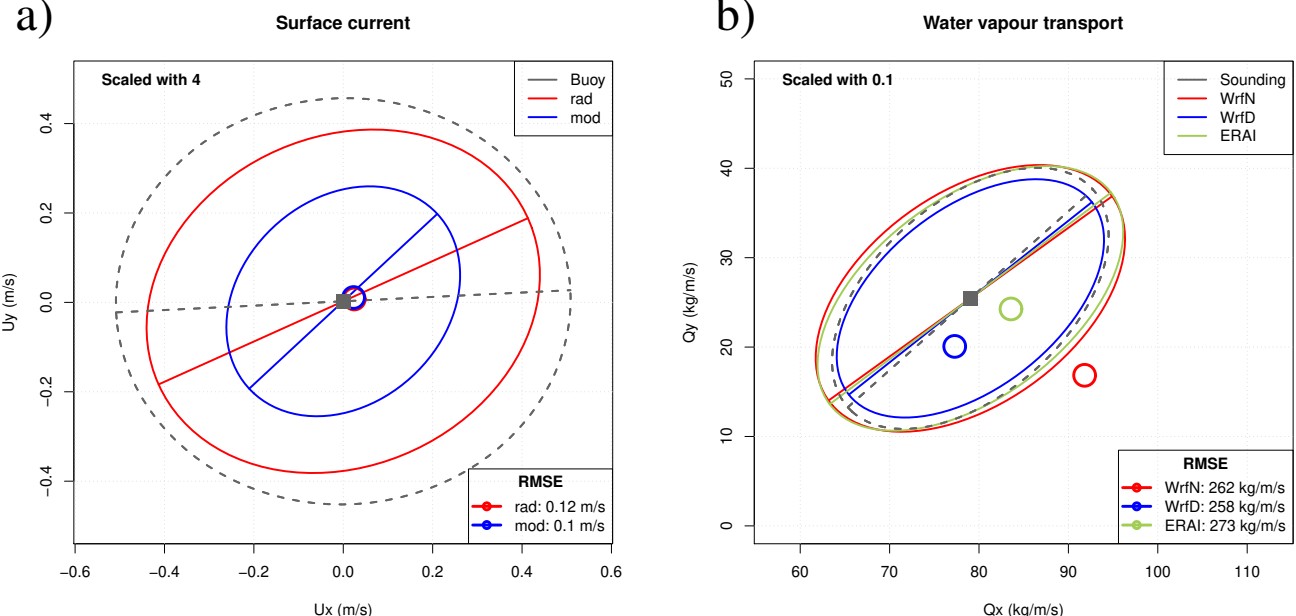

**Figure 6.** Sailor diagram representing the structure of errors between HF radar estimations of currents (rad) and model results (mod) with variances corresponding to EOFs scaled with scale factor of 4 (left). Sailor diagram derived from vertically integrated water vapour transport (right) scaled with a factor of 0.1.

### 4.3    Vertically integrated water vapour transport

The Sailor diagram for the vertically integrated water vapour transport can be seen in Figure 6 (right). In this case, the errors associated to the bias are smaller than the error associated to the covariance. However, since the errors in the anomalies are not very large, the visibility of the diagram has been increased by plotting all of the ellipses on top of the observational point (centered diagram). This way, the errors in direction can be easily identified. For clarity, the ellipses are again scaled with a scale factor of 0.1. It can be seen that the estimation of the EOFs is closer for the case of the simulation with data assimilation, both in direction and, particularly, in the case of the amount of variance represented, since WRF N and ERAI slightly overestimate the water vapour fluxes.

A selection of the tabular results corresponding to the RMSE between observed and modelled vertically integrated water vapour transport are presented in Table 2. Different aspects of the main principal axes such as their semi-major and semi-minor axes, their relative rotation, the two-dimensional correlation coefficient and the combined RMSE can be readily analyzed for the water vapour transport vectors. The two-dimensional correlation coefficient $R^2$ and the RMSE are better for WRF D than for the other models. There is a good agreement in the overall orientation of the leading EOF for all datasets, with the bias being smallest for ERA Interim.

| | Model | $\sigma_x$ | $\sigma_y$ | $R^2$ | $\lvert \bar{\mathbf{U}} - \bar{\mathbf{V}} \rvert$ | RMSE | $\varepsilon$ | $g_{11}$ |
|---|---|---|---|---|---|---|---|---|
| 1 | OBS | 183.45 | 107.83 | | | | 0.81 | |
| 2 | WRF N | 195.53 | 118.21 | 1.57 | 15.41 | 261.98 | 0.80 | 0.99 |
| 3 | WRF D | 173.47 | 100.19 | 1.94 | 5.65 | 257.53 | 0.82 | 1.00 |
| 4 | ERAI | 196.99 | 111.18 | 1.92 | 4.69 | 272.94 | 0.83 | 1.00 |

**Table 2.** Agreement of simulations by different models with observed vertically integrated water vapour transport from soundings. $\sigma_x$ and $\sigma_y$ represent the semi-major and semi-minor axes of the ellipses (kg m$^{-1}$ s$^{-1}$). The $R^2$ column represents the value of the two-dimensional correlation coefficient following Crosby et al. (1993) ($R^2 = 2$ for a perfect model). The differences between the datasets described by the bias $\lvert \bar{\mathbf{U}} - \bar{\mathbf{V}} \rvert$ (kg m$^{-1}$ s$^{-1}$) and total root mean squared error (kg m$^{-1}$ s$^{-1}$) are also shown. Finally, the eccentricity of the ellipses ($\varepsilon$) and the congruence coefficient $g_{11}$ of the EOF1 of every model with the one derived from observations are also shown. The congruence coefficient $g_{11}$ represents the absolute value of the cosine of the relative rotation of model ellipses with respect to the observational one (Section 3.2).

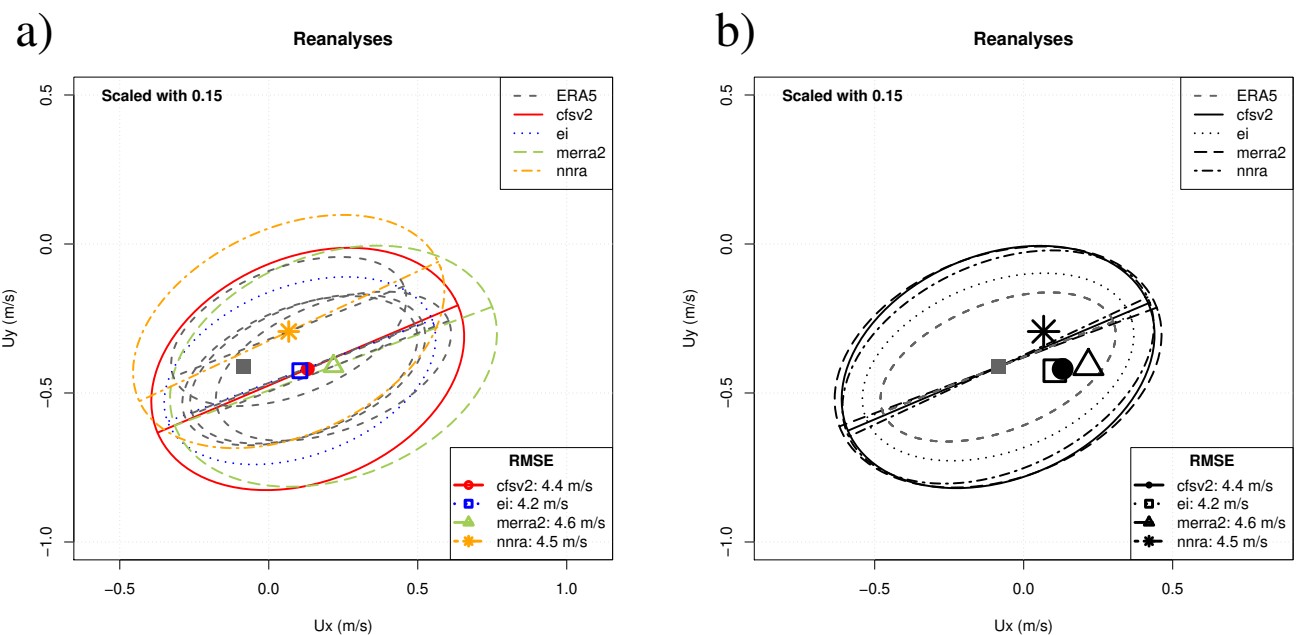

**Figure 7.** Sailor diagram representing the structure of errors in surface wind in January over the Northern Hemisphere for different reanalyses, uncentered version (left, scale factor 0.15) and centered version (right, scale factor 0.15).

## 4.4 Spatial distribution of seasonally-averaged surface wind

As an example of the potential uses of the Sailor diagram, Figure 7 (left, panel a) represents in an uncentered version of the Sailor diagram the agreement of the January-averaged northern Hemisphere surface wind from different reanalyses using an scale factor of 0.15. In order to show that the use of different line styles and colours can lead to diagrams which can be

better interpreted, the left panel is presented with different line-styles. On the other side, Figure 7 (right, panel b) shows the agreement of the January-averaged northern Hemisphere surface wind from different reanalyses using an scale factor of 0.15

in a centered version of the Sailor diagram. In these cases, we are assuming that ERA5 corresponds to the "perfect" dataset (observations). The selection of a reanalysis as a perfect model is quite arbitrary, but we are performing this comparison for the sake of showing the ability of the Sailor diagram to evaluate spatial fields, as was done in the initial design of the Taylor diagram. In this right panel, a black and white version of the diagram is used, to show that it can also be used without different colours, if the linestyle and character used for the reference points are changed. In the black and white version a centered

version of the diagram is used. Since all the ellipses corresponding to the different models are plot on top on the observational average point, the number of ellipses to be used is smaller and the diagram better reflects the directionality problems and the under/over estimations of variances with less lines. It is clearly shown that the reanalyses produced by the ECMWF (ERA5 and ERAI) show the closest agreement both in terms of the smallest bias and better matching of the corresponding EOFs. The other reanalyses (CFSRv2, MERRA2 and NNRA) group along the same semi-major axis, but they overestimate the variability

when compared with ERA5. In terms of the bias, too, it can be seen that the lowest bias is the one corresponding to ERAI. The easiest way to arrive to a numerically precise overall diagnostic is presented in the legend, where the aggregated RMSE error is shown.

## 4.5 Application to multimodel ensembles

In this case, we propose to define the average of all the $M$ ensemble members of every model as the vector $\bar{\mathbf{V}}$ (Rougier, 2016).

On the other side, the principal components and the associated variances and eigenvectors can be estimated from an extended data matrix $\mathbf{V}_e$ (with dimensions $NM \times 2$), which is built by joining all the realizations together in a single dataset. This means that the observational matrix $\mathbf{U}$ must also be extended to an $\mathbf{U}_e$ matrix sized $NM \times 2$. This can be done by repeating the observations $M$ times to produce the $\mathbf{U}_e$ dataset. This ensures that the algorithm will work because the covariance matrices involved will still be of full rank. However, it has to be considered that, in this case, the number of effective degrees of

freedom (Bretherton et al., 1999) in both $\mathbf{U}_e$ and $\mathbf{V}$ datasets will not be the same. This would also be a problem for different models $\mathbf{V}_i$ and $\mathbf{V}_j$, if the number of members in their ensembles are not the same, such as in the CMIP set of runs, for instance.

As shown in Figure 8 (left), prepared using as scale factor 0.2, the Sailor diagram shows interesting features. The three models studied agree quite well in the simulation of the spatial variability of the field (the EOFs and major/minor axes in the ellipse represent the spatial variability of the field). The direction of the EOFs in this case do not represent the physical

direction of wind in the Hemisphere, but the orientation of the leading EOFs. That is, the main axis of spatial variability in the zonal and meridional directions over the Hemisphere (in this case, the diagram represents a time-mean averaged field in a T-mode EOF decomposition). The analysis of the biases shows that both MIROC and IPSL models underestimate zonal average winds when compared with ERA5, whilst HadGEM2-ES shows a slightly higher zonal mean wind. This information can be obtained from the structure of the biases alone. The zonal component of the mean winds, as represented by points (square for

the reference) are close to zero for MIROC and IPSL, but they are positive for ERA5 and HadGEM2-ES. Conversely, MIROC tends to overestimate the mean meridional circulation (red point placed higher than ERA5) and HadGEM2-ES (green point)

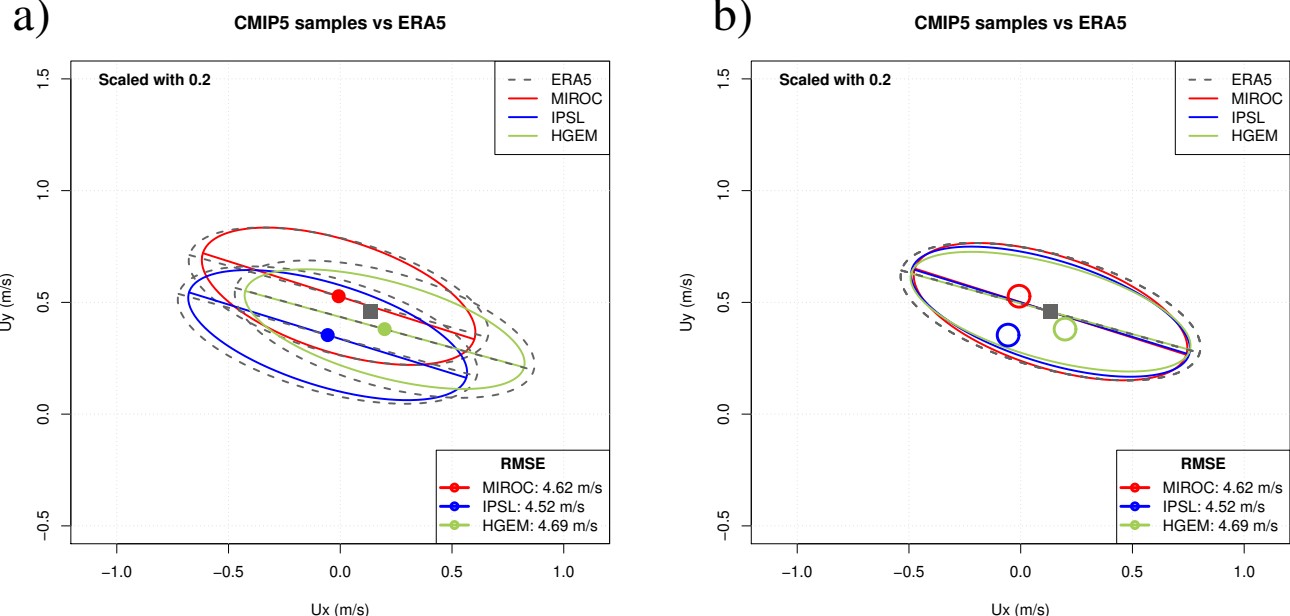

**Figure 8.** Sailor diagram representing the agreement between the Southern Hemisphere wind field as simulated by two models from the CMIP5 repository with ERA5 data when the reference dataset is repeated in an extended matrix. An uncentered version of the diagram (left) is compared with a centered version of the diagram (right).

and IPSL (blue point) underestimate it. In order to show a clearer picture, Figure 8 (right) presents a centered version of the Sailor diagram. The use of centering adds an interesting degree of freedom to the user in order to enhance the visibility of different aspects of the diagram, such as the rotation of the EOFs. For centered diagrams, the ellipses are drawn on top of

the mean hemispheric wind. Thus, only one instance of the observational ellipse is plot. The analysis of the position of the average points leads to the same conclusions regarding the biases as before. The rihgt panel in Figure 8 shows that climate models slightly underestimate the spatial variations of the Southern Hemisphere winds (their semi-major and semi-minor axes are shorter). However, the leading EOF of the spatial variability is very close in all models, as should be expected from the horizontal structure of long-term winds (trades in tropical regions, westerlies in the extratropics). These features are properly

simulated by climate models for the long-term average fields.

The second option for ensembles (same scale factor) is shown in Figure 9 (left). It consists in the use of every single realization of the ensemble as a single model. This case is not of a great scientific interest, but we are presenting it in order to show the behaviour of the diagram with a high number of models (15 realizations in this case). The diagram leads to a neat comparison of the relative performance of the different members of the ensemble. This information might be interesting

because of scientific reasons such as that the initialization of the members of the ensemble uses different techniques which need testing, for instance. In the case shown, the conclusion is quite clear: the averaged bias is relatively independent of the

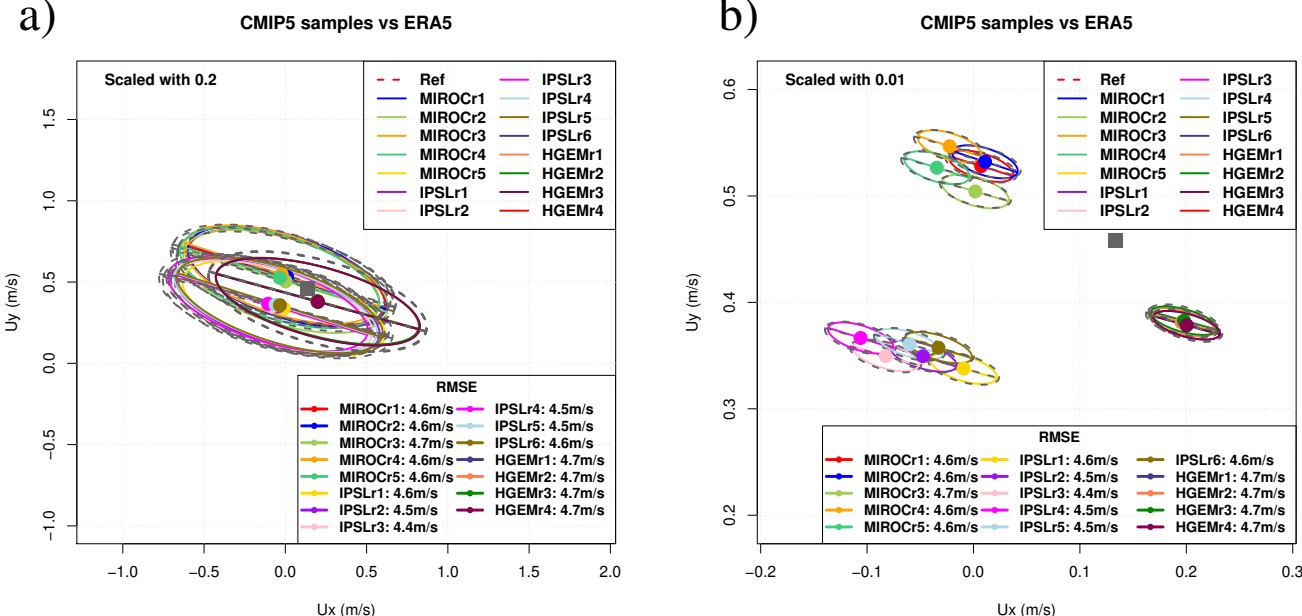

**Figure 9.** Sailor diagram representing the agreement between the Southern Hemisphere wind field as simulated by two models from the CMIP5 repository with ERA5 data when the reference dataset is repeated in an extended matrix (left) or when the individual realizations of the ensemble are taken as independent datasets (right).

realization and the averages corresponding to every model tend to cluster at the same position, with very low biases. The inter-model variability is very low, as could be expected from long-term (27 years) time-averaged fields from climate models. Besides that, the intra-ensemble variance of properties such as the spatial variability of the field is also quite low, so that the

ellipses derived from different realizations in the same model almost completely overlap. Thus, in the analysis performed here, all the realizations of every model in the ensemble are very close to the reference dataset. In order to illustrate the possibility to play with an additional degree of freedom (scale parameter) the right panel of Figure 9 represents the same ellipse with a very small scale factor. It can be seen that all the realizations by HadGEM2-ES are still more or less on top of each other, whilst the ellipses drawn for the realizations from other models start to separate. However, even in this extreme case, the analysis

of the directionality of the leading EOF for fifteen realizations is still possible, since every ellipse can be compared with the reference one corresponding to observations. The legend showing the RMSE supports the conclusion that both the inter-model and intra-model variabilities are very low, as can be expected from a long-term (27 year) averaged Hemispheric wind.

The final decision on the use of one approach (Figure 8) or the other (Figure 9) for the analysis of ensemble integrations is open to the reader, since one or the other will be used by experts to answer different questions, such as whether the internal

variability of the ensemble (in terms of bias and principal directions) is high or low. This might be important in some cases such as operational forecasting, but not in others, such as long-term averaged spatial fields.

## 5   Conclusions

A new diagram for the fast evaluation of the quality of models forecasting two-dimensional vector fields or time series has been presented. As Taylor (2001) properly stated in his seminal paper, a new diagram will only be accepted by users if it helps in the fast and efficient intercomparison of model results against observational datasets. The authors of this paper developed the Sailor diagram in order to fill a gap that we detected when comparing two or more vector fields in our own work. In our previous papers when we worked with with vector fields (Ibarra-Berastegi et al., 2015, 2016) we solved this problem by duplicationg the Taylor diagram, one for each component. The Sailor diagram merges the same information and allows a straightforward visual comparison while rigorously providing the numeric values of the RMSE. It provides additional diagnostics which allow a complete analysis of the errors in the simulated directions too.

The authors hope that the results presented so far demonstrate that the Sailor diagram achieves that goal. First, the diagram relies on the partition of the two-dimensional MSE in its bias and covariance parts. Those two terms are presented in the diagram separately. Thus, those two components of the error can be easily identified for the different datasets. Second, the covariance part is decomposed in terms of the corresponding principal components (empirical orthogonal functions). The structure of the covariance matrix of models and observations can also be effectively compared in the presented diagram, both in terms of the length of their semi-axes (fraction of variance) and in the relative rotation of every model against the reference dataset. This allows to easily identify in the diagram if the models under or overestimate the variance along any of the main axes and whether the main directions of variability in models and observations are relatively rotated or not. Thus, both two-dimensional bias and covariance can be visually identified from the diagram. Since the decomposition of the horizontal vector field is performed by means of two EOFs, there is not loss in the variance of the observed or simulated datasets which are being compared.

The diagram might provide inaccurate estimations of the relative rotations of the principal axes of the distribution of vector components in case both eigenvalues were degenerated and the eigenvectors were affected by substantial sampling uncertainty. In any case, a diagnostic produced by the package we provide, the eccentricity of the ellipses, Eq. (13) can be used by the user to detect this risk. In any case, even if the eigenvalues were degenerated, the final classification of models is performed in terms of the RMSE, which is a measure of error which is not affected by this degeneracy.

The diagram is easily customizable in order to increase the ability to identify features of the datasets being verified by means of the use of scale factors for the ellipses (compare both panels in Figure 5). The diagram can also be customizable by centering all of the ellipses on top of the average corresponding to the reference dataset instead of plotting all of them on top of every model being used. Thus, researchers can design a diagram that best suits their needs. In any case, if the number of models being tested is very high, there will appear many lines, which will make difficult the interpretation of uncentered diagrams. Thus, the option to separately use centered or uncentered diagrams and different scale factors allows to customize the diagram to increase the ability to discriminate between similar biases (use smaller scale factors) or rotation angles (use centered diagrams). In any case, the error scores provided by our implementation (total RMSE, rotation angle, fractions of variance, $R^2$ and many others), as described in Section 3, can also be used in tabular form for a pre-screening of the multimodel dataset. Then, as a final step,

only the most interesting models might be presented. Thus, the combination of centering and scaling strategies and tabular indices as described in Section 4 will lead to an effective verification of vector fields.

The analysis of ensembles can also be performed by means of the diagram. As shown in subsection 4.5, the diagram can accommodate this case by using two different policies. In the first case, all the $M$ members of the ensemble belonging to a single model can be mixed in a unique dataset, but this involves repeating the block of observations $M$ times (Figure 8). This

implies that the analysis of the results presented in the diagram in this case must consider the different number of effective degrees of freedom very carefully and further research should be performed to analyze the impact of this in the application of the Sailor diagram to model ensembles. However, in the second case, all the ensemble members are analyzed as independent realizations of the same dataset (Figure 9). This tends to clutter the diagram, but these results are not affected by problems related to the number of effective degrees of freedom in the different datasets used to build the diagram. The decision on the

use of one or the other depends on the application intended by the user.

As a conclusion, we hope that the diagram presented here, together with an R implementation of it freely available in CRAN will ease the verification of vector fields derived from geoscientific models in the future.

*Code and data availability.* The code used to prepare the figures in this paper is described as examples in the manual of the R package `SailoR`, available from CRAN https://cran.r-project.org/package=SailoR. The data used to produce these figures are also distributed with

the package. The version of the package used to prepare the figures in this paper can be found in https://doi.org/10.5281/zenodo.3839933.

*Author contributions.* JS conceived the idea, most of the mathematical analysis and wrote some parts of the linear algebra code and most of the paper. SCM collaborated in the analysis of the matricial structure of the error and wrote substantial parts of the code, particularly the graphical representation of data. GE collaborated in the preparation and testing of the linear algebra part and provided data for the tests. SJGR prepared the R package distributed with the paper and its documentation. SCM, GIB and AU provided data for the package, performed

exhaustive checking of the implementation and helped in the analysis of results. All authors took active part in the writing of the paper.

*Competing interests.* The authors manifest they have no competing interests in the outcome of this paper-edition process.

*Disclaimer.* The code is made publicly available without any warranty.

*Acknowledgements.* This work has been funded by the Spanish Government's MINECO project CGL2016-76561-R (AEI/FEDER EU) and the University of the Basque Country (UPV/EHU funded project GIU17/02). The ECMWF ERA-Interim data used in this study have been

obtained from the ECMWF-MARS Data Server. The authors wish to express their gratitude to the Spanish Port Authorities (Puertos del

Estado) and Basque Meteorological Agency (Euskalmet) for being kind enough to provide data for this study and for allowing us to make the data publicly available in the `SailoR` package. CMIP5 model output data provided by ESGF have been used for this paper. Constructive comments by three anonymous reviewers have lead to a better version of the manuscript.

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
