# Peer review of "The Sailor diagram. A new diagram for the verification of two-dimensional vector data from multiple models."

_Geoscientific Model Development, 2019_

## Referee Comment (RC1) · Anonymous Referee #1 · 3 Jan 2020

The paper addresses a relevant and often appearing issue: comparing vector quantities. It reviews the different approaches developed so far, giving appropriate credit to those, and adds the idea of a novel graphics presentation as "sailor diagram". This is potentially a useful tool for a vast range of applications, several examples are chosen from different fields for illustration. The deviation of method is clearly outlined and valid, reproducibility is excellent. The title is excellently chosen, abstract is concise and the term "sailor diagram" justified in the paper. Language and maths are clear. Figures are less clear. Grey squares in all figures are hard to spot (and important). Although it is nice that the figures relate to real world examples, for introducing the concept it would be helpful to have figures showing clearly the benefits and limitations of the sailor diagram. Figure 2a is a very good one. The others are not easy to interpret, i.e.,

helping less to understand the concept. Applicability and interpretation, and general presentation would benefit from clearer examples. Given that the graphics are a central idea of the paper, following revisions are suggested, with the intention to improve understanding and uptake of the Sailor diagram for other researchers:

1) List and number the features of the Sailor diagram clearly, eg., like i) size of ellipse depicting covariance ii) direction of ellipse indicating error main axis iii) squares indicating bias for both components iv) options for scaling as indicated in Fig. 1

2) give one (possibly synthetic) example figure illustrating clearly each feature (e.g., datasets disagreeing on i) and agreeing on ii) and iii). For iv) note what scaling comes with which advantage / disadvantage.

3) Explain the underlying assumptions and the limitations (i.e., what could go wrong with the interpretation). For instance, in Fig 1, the almost orthogonal major axes – are they caused by the two EOF being approx. same size and some noise deciding on whether the correct EOFS are aligned in the graphics? Are Fig. 1 (major and minor axis) thus showing a possible pitfall of interpretation of the Sailor diagram? What other limitations and possible pitfalls do exist ?

4) Remove figures not adding information. The whole section 2 (data description) is not necessary for the understanding of the principle of the Sailor diagram and can be shortened significantly, just serving the understanding of real world examples. It is not clear for what Fig. 3a is needed – and its explanation is full of abbreviations (check "per" and "pers"). Somebody not familiar with these particular data sets cannot extract sensible information from section 4.4.

5) Figure 4 (right) needs clarification. It is impossible to relate the color codes to the 2 clusters of ellipses. Why are there exactly 2 clusters of ellipses? Furthermore, it is unclear what the centres should denote. Why are there 2 grey ellipses in the upper cluster? It is unclear what is intended to show. I cannot draw conclusions from this figure. Either clarify or remove this figure.

6) It is commendable you provided an R package SailoR. It would be good to state clearly in section 3.7 which figure is included in the manual (instead "some of these plots")

7) For better visibility, consider plotting the squares on top of the lines, to change grey to black, and to enlarge the size of the squares.

---

## Referee Comment (RC2) · Anonymous Referee #2 · 13 Jan 2020

Comments on the manuscript entitled "The Sailor diagram. An extension of Taylor's diagram to two-dimensional vector data" by Sáenz et al. submitted to GMD

Recommendation: Major revision

Summary comment The authors aim to propose a model evaluation method in terms of vector data. They constructed a "Sailor diagram" and claimed that this diagram is an extension of Taylor diagram. In my point of view, it is very farfetched to say the Sailor diagram is an extension of Taylor diagram. The Sailor diagram is not even like Taylor diagram. Two diagrams presents very different statistics. For example, Taylor diagram can illustrate correlation coefficient, standard deviation, and RMSE. However, the Sailor diagram shows the first and second EOF with the semi-major and semi-minor axes of ellipses, respectively. Each ellipse represents one model or observational data, the difference between model and observation is judged visually, which is less objective. More importantly, why are the first and second EOFs useful and what is the implication of the EOFs in terms of climate model evaluation? The EOFs between model and observation may represent different patterns. In this case, the comparison between model and observation can give wrong conclusions. These questions were not clearly interpreted (addressed) in the manuscript. Substantial revision is needed before the manuscript can be considered for publication in GMD. Detailed comments are listed below.

1. The title of the manuscript is misleading and should be changed because the Sailor diagram is totally different from the Taylor diagram. Two diagrams present very different statistics and have different implications. For example, Taylor diagram presents correlation coefficient, standard deviation, and RMSE. However, the Sailor diagram was constructed based on the EOF of vector data, which does not explicitly include correlation coefficient and standard deviation. Two diagrams do not look like each other, either.

2. Section 2 introduced five different vector datasets using 3 pages. It's not necessary to use so much dataset and can be reduced since they are all vector data. Only one or two of them should be enough to interpret the diagram. In contrast, methodology (section 3) is the key part of the manuscript which should clearly interpret and explain the method. However, the methodology was not well interpreted and hard to follow. I suggest that the authors interpret the methodology using an example data. Eu, Pu, and EOFs can be illustrated by using the example data to help readers to understand the method. The method and its application should be clearly interpreted in terms of model evaluation. In addition, section 3.1 and 3.2 generally present the same equations and can be merged.

3. Line 49-51 and 78-80: This is not true. The Taylor diagram can be extended to two (even more) dimensional vector data evaluation directly by using a set of statistical quantities defined by Xu et al. (2016). This paper was also cited by the authors.

4. Line 99-101, 109-112: To my knowledge, Xu et al. (2016) did normalize various statistics but no approximation was applied. The author argued that the merit of the Sailor diagram is that no approximation is needed. However, Sailor diagram illustrate the first two EOFs. Each EOF can only explain part of the variability of the original data.

5. Line 220-223: It is confusing that the authors use "U" to represent observation and "V" to represent model because U and V were usually used as the zonal and meridional component of wind. I suggest the authors replace "U" with "O" and "V" with "M" or other appropriate notation to avoid confusion.

6. Line234: How are the principal components of the data standardized?

7. Line 360-361: What is the implication of the relative rotation between EOFs from observations and simulations? Why is it important to model evaluation?

8. Line 361-363: Why the variance explained by each EOF is important in terms of model evaluation? What if the EOFs between model and observation represent different patterns?

---

## Author Comment (AC1) · 13 Jan 2020

*Our reply to the original comments by the reviewer is written in italics.*

The paper addresses a relevant and often appearing issue: comparing vector quantities. It reviews the different approaches developed so far, giving appropriate credit to those, and adds the idea of a novel graphics presentation as "sailor diagram". This is potentially a useful tool for a vast range of applications, several examples are chosen from different fields for illustration. The deviation of method is clearly outlined and valid, reproducibility is excellent. The title is excellently chosen, abstract is concise and the term "sailor diagram" justified in the paper. Language and maths are clear.

*Thank you for your kind comments.*

[Figure]

Figures are less clear. Grey squares in all figures are hard to spot (and important).

*We appreciate this comment by Reviewer 1 and we agree with his/her appreciation. We are reworking the package so that in a new version these grey squares will be larger and they will be on top of all the lines.*

Although it is nice that the figures relate to real world examples, for introducing the concept it would be helpful to have figures showing clearly the benefits and limitations of the sailor diagram. Figure 2a is a very good one. The others are not easy to interpret, i.e., helping less to understand the concept. Applicability and interpretation, and general presentation would benefit from clearer examples. Given that the graphics are a central idea of the paper, following revisions are suggested, with the intention to improve understanding and uptake of the Sailor diagram for other researchers: 1) List and number the features of the Sailor diagram clearly, eg., like i) size of ellipse depicting covariance ii) direction of ellipse indicating error main axis iii) squares indicating bias for both components iv) options for scaling as indicated in Fig. 12) give one (possibly synthetic) example figure illustrating clearly each feature (e.g.,datasets disagreeing on i) and agreeing on ii) and iii). For iv) note what scaling comes with which advantage / disadvantage.

*We again appreciate this constructive comment by Reviewer 1. We find it a very useful suggestion. We have prepared a new figure (it will be Figure 1 in the revised version of the manuscript). In it, we have taken a real-world example (one year -2018- of hourly wind data in a grid point in front of Los Angeles from ERA5, 38 degrees North and 124 degrees West). These data are considered as the reference dataset. From this reference dataset, new synthetic datasets with artificial sources of error are produced. In the first case (MOD1), a constant bias is added to the reference dataset. Thus, the orientation and lengths of MOD1 are the same as the ones from the reference dataset, but the bias is different from zero. This addresses source of error iii, as indicated by the reviewer.*

*A second model (MOD2) is generated by rotating 30° counter-clockwise the wind fields. This introduces an artificial rotation of the EOFs without affecting the fractions of variance of the principal axes. The rotation changes the position of the average wind and it also introduces some bias. This second synthetic dataset addresses sources of error iii (bias) and ii (rotation), as suggested by reviewer.*

*As a completely artificial example, we have also generated model MOD3, which only involves a random resampling in time of wind measurements. Thus, the bias source of error must be zero and the EOFs must also be the same, but due to the fact that the correlation coefficient between datasets now is not the same, in this case the sailor diagram shows a perfect graphical agreement, but the legend shows that the RMSE is not zero, so that this part is due to the lower correlation of the wind. This case addresses the lack of temporal correlation of the data despite perfect agreement in length of major axes of the ellipse (case i suggested by reviewer), angle (case ii) and no bias (case iii).*

*The last synthetic case (MOD4) is produced by multiplying the original data by a scale factor (2), so that the axes of the ellipses are changed (case i suggested by reviewer), but without any rotation of the axes and a different bias due to the scaling factor. We have produced versions centered and uncentered of the Sailor plot for these synthetic datasets and we intend to use this as the first Figure of the paper so as to address this interesting point raised by the reviewer. We agree with him/her that this addition will improve the interpretability of the paper. We enclose the new figure below.*

*Figure 1: Sailor diagrams representing each of the synthetic datasets including the artificial sources of errors described above in the text. Two versions are included, one uncentered (left) and another one where all the ellipses are centered over the reference value (right).*

*These considerations are clearly reflected in the values of the table exported by the package (call to function SailoR.Table), as shown below. This table will also be added*

*to the last version of the paper as Table 1.*

3) Explain the underlying assumptions and the limitations (i.e., what could go wrong with the interpretation). For instance, in Fig 1, the almost orthogonal major axes – are they caused by the two EOF being approx. same size and some noise deciding on whether the correct EOFS are aligned in the graphics?

*The reviewer is right that in this case, the standard deviation of the reference dataset seems the same along both EOFs, despite it is not exactly the same (4.3 m/s versus 4.2 m/s). For the rest of models, it is not so close (4.21/3.62, 4.75/4.29, 3.68/3.47, 4.23/3.99). Therefore, we are sure the results are not purely due to degeneracy of the eigenvalues of the covariance matrix in this case. The reviewer raises a valid point in this comment, and we will modify the package to check this case, raising a warning for the user in the next release of the package. However, in our experience, this will be more the exception than the norm for realistic vector time series.*

Are Fig. 1 (major and minor axis) thus showing a possible pitfall of interpretation of the Sailor diagram? What other limitations and possible pitfalls do exist?

*Besides the fact that the Sailor diagram presents in graphical terms the bias and the EOFs, the legend at the lower-right corner of the diagram shows also the RMSE of the time series. Even though the EOFs were exactly the same (see MOD3 in Figure 1 in this response), the RMSE legend, which presents in an aggregated index the amount of the error due both to the bias and the EOFs would still point to the dataset showing the smallest error. We will stress this point in the final version of the manuscript.*

4) Remove figures not adding information. The whole section 2 (data description) is not necessary for the understanding of the principle of the Sailor diagram and can be shortened significantly, just serving the understanding of real world examples.

*We tried to thoroughly describe all the datasets used to ease replicability, but we will shorten this section in the last version of the manuscript following the suggestion by*

*the reviewer.*

It is not clear for what Fig. 3a is needed – and its explanation is full of abbreviations (check "per" and "pers"). Somebody not familiar with these particular data sets cannot extract sensible information from section 4.4.

*We intended to show that the SailoR diagram, besides playing a role in the evaluation of typical vector quantities such as wind or current can also be used for additional vector quantities, such as wave energy flux. In this particular case, we find interesting that the EOFs from techniques which involve the resampling of the original dataset such as persistence and analogs ("pers", "analo" and "analorf") show the smallest rotation of the EOFs. As we mentioned above in this reply, the information about the RMSE in the bottom-right legend shows that the combination of bias and EOFs leads to smaller RMSE in the case of the "analo" dataset. We would prefer to keep this figure, because we think it is interesting, but we agree with the reviewer that it needs a better explanation along the lines written in this reply. However, we could also remove it, in case the reviewer still thinks it must be removed.*

5) Figure 4 (right) needs clarification. It is impossible to relate the color codes to the 2 clusters of ellipses. Why are there exactly 2 clusters of ellipses? Furthermore, it is unclear what the centres should denote. Why are there 2 grey ellipses in the upper cluster? It is unclear what is intended to show. I cannot draw conclusions from this figure. Either clarify or remove this figure.

*We understand the comments by the reviewer. In this case, again, we find the Figure should be kept in the final version of the manuscript, since it illustrates one of the problems of one of the ways that the Sailor diagram can be used to assess ensembles.*

*Regarding color codes, we can change the color palette, but it will be of no help, and the reason is that the ellipses are centered almost at the same point (but not exactly at the same point) for each of the members of the ensemble (clusters defined by realizations of the same model). In this case, the figure shows that the bias of each of the*

*realizations by each of the models is quite the same (but not exactly the same). Since this figure is an uncentered version of the Sailor diagram, every realization (that are being taken as independent simulations in this case) implies that there is a grey ellipse centered at every mean value. The intra-model bias is much smaller than the inter-model one, and that is the reason that there are two clusters of ellipses (one cluster involving all the realizations from MIROC and another cluster involving all the realiza-tions by IPSL). The centre of each ellipse denotes the mean of every realization, and since they are very close, it seems that they are the same, but they are not. We add here a centered version of the plot which could also be used. However, we, as authors, would prefer the original one since, in the case of the Figure 2 below, the individual realizations can be barely identified. Figure 2: Centered version of Figure 4 (right) of the manuscript.*

6) It is commendable you provided an R package SailoR. It would be good to state clearly in section 3.7 which figure is included in the manual (instead "some of these plots")

*We will clarify this in the final version of the manuscript and the new version of the manual, when we upload the modified version of the package considering all the suggestions made by reviewers.*

7) For better visibility, consider plotting the squares on top of the lines, to change grey to black, and to enlarge the size of the squares.

---

## Author Comment (AC2) · 13 Jan 2020

This is the new figure 1 we propose
* * *
[Figure]

[Figure]

**Fig. 1.**

---

## Author Comment (AC3) · 13 Jan 2020

New Table 1 we propose

| | Model | $\sigma_{Ux}$ | $\sigma_{Uy}$ | $\sigma_{Vx}$ | $\sigma_{Vy}$ | $\theta_u$ | $\theta_v$ | $\theta_{vu}$ | $R^2$ | \|bias\| | RMSE |
|---|-------|---------------|---------------|---------------|---------------|------------|------------|---------------|-------|----------|------|
| 1 | Ref   | 3.27 | 6.07 |      |       | 1.93 |      |       | 2.00 | 0.00 | 0.00 |
| 2 | MOD1  |      |      | 3.27 | 6.07  |      | 1.93 | -0.00 | 2.00 | 8.34 | 5.56 |
| 3 | MOD2  |      |      | 5.22 | 4.51  |      | 2.46 | 0.52  | 2.00 | 2.88 | 8.69 |
| 4 | MOD3  |      |      | 3.27 | 6.07  |      | 1.93 | -0.00 | 0.00 | 0.00 | 1.84 |
| 5 | MOD4  |      |      | 6.54 | 12.14 |      | 1.93 | 0.00  | 2.00 | 5.56 | 11.76 |

**Fig. 1.**

---

## Author Comment (AC4) · 13 Jan 2020

This could be a different figure for right panel in Figure 4, but we prefer to keep the current version

[Figure]

**Fig. 1.**

---

## Author Comment (AC5) · 20 Jan 2020

**Reply to interactive comment by reviewer 2 on "The Sailor diagram. An extension of Taylor's diagram to two-dimensional vector data" by Jon Sáenz et al.**

Note by the authors: The original text by reviewer is written in **boldface**

**Summary comment The authors aim to propose a model evaluation method in terms of vector data. They constructed a "Sailor diagram" and claimed that this diagram is an extension of Taylor diagram. In my point of view, it is very farfetched to say the Sailor diagram is an extension of Taylor diagram. The Sailor diagram is not even like Taylor diagram. Two diagrams presents very different statistics. For example, Taylor diagram can illustrate correlation coefficient, standard deviation, and RMSE. However, the Sailor diagram shows the first and second EOF with the semi-major and semi-minor axes of ellipses, respectively.**

Thank you for your comment. We understand that our work needs some clarification. Please note that:

1. All the Sailor diagrams presented in our paper show the RMSE of the vector timeseries or fields in a legend. The RMSE for two-dimensional vector data can not simply be presented as a plot if at the same time we keep the relevant information regarding the difference in orientation of the principal axes. Thus, the Sailor diagram presents the RMSE but,

additionally, it allows to identify errors in the directions and fractions of variance in models and observations.

2. Regarding the mention to correlation by reviewer 2, as we show in our manuscript (section 1, particularly page 3), there does not exist a unique definition of the correlation for two-dimensional time series. See references in our manuscript (Cramer, 1974; Crossby et al., 1993; Jupp and Mardia, 1980; Robert et al., 1985; Stephens, 1979) for further details. However, the two-dimensional correlation coefficient based in canonical correlations, which is the most widely used in the literature, is presented in Table 1 of the manuscript and can be computed with the package we present. It could also be added to the diagram by means of additional ellipses or lines, but after some previous tests at the preliminary stages, we decided to remove them, since in our view, it would not help in the interpretation of the results.

3. Regarding the semi-major and semi-minor axes, we think it is not a weakness of the diagram, but a powerful diagnostic tool in it, instead. We find a very useful contribution of this diagram the fact that the standard deviations of models and observations can be compared visually from the comparison of the

axes of the ellipses in the diagrams. These values can also be retrieved as numbers from the implementation we have developed as a R package. Thus, regarding the mention by the reviewer that we do not present standard deviations, our answer is that we present the standard deviations of the major and minor axes of the distribution of the vector field in two orthogonal directions. This is better (in our opinion) that just adding both standard deviations in a number. For us, this is a strong positive characteristic in our diagram.

4. Regarding the statement by the reviewer that our diagram does not follow the design of the Taylor diagram, we clearly stated that in the initial submission of our paper (lines 104-106): *we have decided to follow a new approximation which does not lead to the common Taylor diagram used for scalars, but gives more information about the structure of the two-dimensional errors*. Thus, we are not closely following the design of the Taylor diagram because we preferred to present the information related to errors in the direction of vectors. This information about the directionality of the vectors can not be identified in a Taylor diagram designed for scalars.

Considering the previous comments above, we think that the

Sailor diagram represents an important contribution since, to the best of our knowledge, this diagram is the only one which allows to make a full assessment of errors in the orientations and lengths of the semi-major and semi-minor axes of the horizontal distribution of vectors. We are not aware of any alternative readily available tool like the one we have developed to implement the comparison of vector fields *considering them as vectors*. It is a diagnostic tool which provides a very good capability to visually and easily compare the bias, the main directions of variability of the horizontal vectors and their relative variances.

**Each ellipse represents one model or observational data, the difference between model and observation is judged visually, which is less objective.**

The fundamental idea behind a diagram such as the Taylor diagram or the Sailor diagram is that they are designed to easily (visually) show the relative benefits (and also weaknesses) of different models against observations. As such, the diagram we present is designed (as it was the Taylor diagram) to allow this visual comparison, as we assert in our manuscript. However,

the use of the exact RMSE errors in the legend of the diagrams in the manuscript allows a completely objective comparison of model data and observations, since the RMSE data contains an aggregated estimation of the errors due to bias and principal components as well. We carefully partition the sources of RMSE errors in bias, rotation and differences in variances. The paper presents the corresponding equations, and all of them are presented in a diagram. Besides that, they can also be retrieved numerically from the R-package we provide.

**More importantly, why are the first and second EOFs useful and what is the implication of the EOFs in terms of climate model evaluation? The EOFs between model and observation may represent different patterns. In this case, the comparison between model and observation can give wrong conclusions. These questions were not clearly interpreted (addressed) in the manuscript.**

We find the EOFs are fundamental in the identification of the errors in direction between modelled and observed winds. Perhaps the reviewer is considering here that we are applying EOFs in the

traditional space-time decomposition of fields as follows:

$$A\left(\mathbf{r}, t\right) = \bar{A} + \sum_k \lambda_k p_k\left(t\right) A_k\left(\mathbf{r}\right),$$

where $\mathbf{r}$ represents the grid points, $p_k\left(t\right)$ the $k$–eth principal component, $\lambda_k$ the $k$–eth standard deviation and $A_k\left(\mathbf{r}\right)$ the $k$–eth empirical orthogonal function. However, in our methodology, we are just using EOFs to identify the main directions of variability of the two-dimensional time-series/spatial fields in order to be able to describe the matching in direction of model data with observations.

The enclosed figure explains with the help of a scatterplot the derivation of the ellipses and their relationship with EOF analysis and the standard deviations mentioned by the reviewer before. We think that, if the editor and reviewers agree, it could be added to the manuscript as Figure 1 since this was also requested by Reviewer 1. It would help in the interpretation of the paper, and it would lead to a better version of it.

In this figure, panel top left (a) presents the scatterplot which can be constructed with one year (2018) of surface wind in front of Los Angeles. It is the same reference dataset we have prepared for Reviewer 1. The red ellipse corresponds to the eigenvectors (major and minor axes, matrix $\mathbf{E}_u$ in equation 5 of our original

submission) and eigenvalues (semiaxes of the ellipses, matrix $\Sigma_u$ in equation 4 in our initial submission) from a EOF decomposition of the two-dimensional covariance matrix, computed from the zonal and meridional components of the observed wind (reference dataset, $\mathbf{U}$). The red point represents the mean of the wind ($\bar{\mathbf{U}}$ in our submission, zonal and meridional components), also indicated by the horizontal and vertical lines passing through it. The ellipse represents the conic section expressed by equation (2) in our submission.

The top-right panel (b) presents the same representation for the reference dataset together with a similar scatterplot (grey colour) representing model MOD1 that we prepared in our answer to Reviewer 1 by adding a constant bias $\mathbf{b} = (4.8, -6.8)$ m/s. The dark brown ellipse represents the major axes of variability of MOD1, their standard deviations centered on its mean (elements $\mathbf{E}_v$, $\Sigma_v$ and $\mathbf{V}$ in our submission). Since MOD1 only involves the addition of a constant bias, both ellipses are of the same dimensions and are oriented similarly. Panel c) shows in a similar way (scatterplot plus ellipses) a comparison of the reference dataset (black) and a second model MOD2 (grey) which has been calculated by rotating the observed winds counter-clockwise by $30°$. The brown ellipse shows that the major and minor axes of the ellipse (eigenvectors)

are rotated accordingly. However, their standard deviations are the same, because the difference between both datasets is limited to an orthonormal transformation (rotation). Panel d shows the behaviour of the data in model MOD3, in which a resampling in time of the same wind vectors as the ones in the reference dataset is performed. By means of this operation, the average wind doesn't change, and the major and minor axes are also the same. However, the correlation of the zonal/meridional components of wind must be close to zero. These results are correctly represented in the Sailor diagram, when the RMSE component is reported. Panel e represents a comparison of the reference data and model MOD4, which has been built by multiplying the reference data by 2. This implies that the average changes and the standard deviation along the major and minor axes doubles, as correctly represented in the diagram. Panel f is similar to the sailor diagram in the sense that the previously shown scatterplots are removed in order to allow an easy comparison of the main components of the errors (bias, rotation and standard deviations along the major and minor axes). Centered and uncentered versions of the Sailor diagram for these synthetic datasets are shown in the reply to Reviewer 1. Authors appreciate this constructive suggestion by both reviewers (build an easy example to illustrate the diagram), since

it will lead to a better paper.

We hope that this new figure clarifies the way we compute the EOFs and the important role played by these ellipses in the Sailor diagram. They represent the standard deviations along the main orthogonal directions of variability of the horizontal vectors.

**Substantial revision is needed before the manuscript can be considered for publication in GMD. Detailed comments are listed below.**

**1. The title of the manuscript is misleading and should be changed because the Sailor diagram is totally different from the Taylor diagram. Two diagrams present very different statistics and have different implications. For example, Taylor diagram presents correlation coefficient, standard deviation, and RMSE. However, the Sailor diagram was constructed based on the EOF of vector data, which does not explicitly include correlation coefficient and standard deviation. Two diagrams do not look like each other, either.**

As we stated before, the Sailor diagram effectively shows the

standard deviation of each dataset (axes of ellipses) and the RMSE in the legend of the plot. However, it can not show the correlation because there is not a universally accepted definition of correlation in two dimensions. Besides that, it also shows the errors in direction and the bias component of the error. Thus, we find it is a very efficient and objective way of comparing model results to observations. Thus, we find "Sailor diagram" is a good way of defining it, as also acknowledged by Reviewer 1. We insist in keeping this name (easy to remember, much better than any acronym we could imagine, and culturally neutral). Besides that, in our view, for the sake of coherence, the name sailor must be kept, since the sailoR package[1] that we distribute in CRAN is also called that way.

**2. Section 2 introduced five different vector datasets using 3 pages. It's not necessary to use so much dataset and can be reduced since they are all vector data. Only one or two of them should be enough to interpret the diagram.**

Thanks for this suggestion. We have decided to reduce the
* * *
[1] https://cran.r-project.org/package=SailoR

number of observational datasets included in the final version of the manuscript since both reviewers agreed on this. Our intention was to show that the diagram can be applied for many different variables in many different fields of study.

**In contrast, methodology (section 3) is the key part of the manuscript which should clearly interpret and explain the method. However, the methodology was not well interpreted and hard to follow.**

We are a little bit surprised by this comment, since Reviewer 1 found the description of the methodology very clear. However, we hope that the addition of Figure 1 as shown in this reply will improve the understanding of the methodology. It will also be added in the final version of the manuscript if the editors and the reviewers agree on including it there.

**I suggest that the authors interpret the methodology using an example data. Eu, Pu, and EOFs can be illustrated by using the example data to help readers to understand the method.**

We agree with the reviewer in this point, and as we already showed in the reply to Reviewer 1, we will include an initial figure built by using synthetic datasets with the aim of describing the main characteristics of the diagram. The Figure that we have prepared for this reply would be Figure 1 in this document. It is an extension of the one we prepared in the answer to Revgiewer 1's comments. Thus, we hope that the methodology is clearly explained now and that we meet the requirements by both reviewers with this new figure.

**The method and its application should be clearly interpreted in terms of model evaluation. In addition, section 3.1 and 3.2 generally present the same equations and can be merged.**

We accept that section 3.2 it is a little bit repetitive, but we found it was interesting, particularly because section 3.3 is an important part of the description of the diagram, as it justifies the errors due to rotation of the model data with respect to observations. We will make an effort in making it shorter without penalizing the interpretation of the methodology. Removing it completely

would probably make difficult to understand the role played by the relative rotation matrices, so we would prefer not to remove it completely from the manuscript. At least some of the equations must be kept.

**3. Line 49-51 and 78-80: This is not true. The Taylor diagram can be extended to two (even more) dimensional vector data evaluation directly by using a set of statistical quantities defined by Xu et al. (2016). This paper was also cited by the authors. Line 99-101, 109-112: To my knowledge, Xu et al. (2016) did normalize various statistics but no approximation was applied.**

It was not our intention to demean Xu et al. (2016) paper when we mentioned the word "approximation". In fact, we also used the word "clever" when we referred to it. We just noticed that in some cases (see, for example their equation 9 or their section 3), the Cauchy-Schwartz inequality was used. That means that the norms of some of the vectors presented are actually upper bounds of the true norms and equation 9 is a good example. But we insist that it was not our aim to demean that paper, so that

in the final submission we will just remove these mentions to the paper by Xu et al., which we consider a very good paper.

**The author argued that the merit of the Sailor diagram is that no approximation is needed. However, Sailor diagram illustrate the first two EOFs. Each EOF can only explain part of the variability of the original data.**

As we wrote in lines 241 and 266 of the initial submission, the covariance matrix that we use is the one built using the zonal/meridional components at each time-series or the spatial distribution of an averaged wind field. This is hopefully better illustrated in Figure 1 added to this reply. Thus, this covariance matrix is a rank 2 matrix. The only exception to this would be the case of a completely degenerated and physically unrealistic flow (laminar). Therefore, the covariance matrix in equations 4 and 12 is representing a full-rank matrix for all sensible cases. We hope that the interpretation will be clear now with the addition of Figure 1 .

**5. Line 220-223: It is confusing that the authors use "U" to**

[Figure]

**represent observation and "V" to represent model because U and V were usually used as the zonal and meridional component of wind. I suggest the authors replace "U" with "O" and "V" with "M" or other appropriate notation to avoid confusion.**

We, sincerely, do not see any possible confusion, as the matrices $\mathbf{U}$ and $\mathbf{V}$ are typed using bold font. They are, thus, matrices, and the text clearly states their dimensions (rows and columns). We would prefer to keep the current notation, which has been defined as clear by Reviewer 1. Besides that, the current notation is the one used in the R package for the Sailor diagram already distributed by CRAN. This way, the paper using the current notation serves as an additional documentation file for the package. However, if we receive any indication from the editor indicated that we must change the notation, we will do it.

**6. Line234: How are the principal components of the data standardized?**

They are not standardized. The fact that they are not standardized allows us to make a full analysis of the RMSE error of the original fields. Perhaps we didn't make that clear enough, but we will clarify it in the final version of the manuscript by explicitly asserting they are not standardized.

**7. Line 360-361: What is the implication of the relative rotation between EOFs from observations and simulations? Why is it important to model evaluation?**

The EOFs of an anemometer/vane represent an orthogonal basis in the horizontal plane. If this basis is different for model and observations, this difference means that the distribution of the horizontal wind in the zonal/meridional plane from model and observations is different (see the case of the Reference dataset and MOD2 in the Figure enclosed to this reply). Thus, changes in rotation of the EOFs imply that there is an error in the directionality of the simulated data. The reviewer has to keep in mind that we are applying the EOFs to the time-series (or spatial distribution) of a 2x2 covariance matrix. Thus, the spatial/temporal variability of the field is not being analysed. We hope this is clear now with the new figure we provide.

**8. Line 361-363: Why the variance explained by each EOF is important in terms of model evaluation?**

It is important because the horizontal distribution of the zonal/meridional components in the horizontal plane defined by the zonal and meridional components must be as close as possible for the simulated wind fields. We stress again that our EOFs analyse the structure of a two-dimensional covariance matrix.

**What if the EOFs between model and observation represent different patterns?**

In that case, the agreement in terms of the RMSE (as described by equation 22) will be lower, as correctly shown in Figures 3b and 4 from the manuscript. We stress again that our EOFs are computed in a two-dimensional covariance matrix both for model data and for observations. We hope that the new figure makes this clear.

[Figure]

2019.

[Figure]

**Fig. 1.**

---

## Author Response (AR1)

Dear Editor.

Please, find enclosed the revised version of original manuscript "**The Sailor diagram. An extension of Taylor's diagram to two-dimensional vector data**" by J. Sáenz, S. Carreno-Madinabeitia, G. Esnaola, S. J. González-Rojí, G. Ibarra-Berastegi and A. Ulazia, https://doi.org/10.5194/gmd-2019-289

Together with this cover letter, we are submitting the revised manuscript, a careful rebuttal to all points raised by both reviewers and a PDF with the changes introduced to the manuscript properly highlighted.

First, we would like to draw your attention to the fact that, because of the requirement by Reviewer #2, we have changed the title of the paper to "**The Sailor diagram. A new diagram for the verification of two-dimensional vector data from multiple models**". In any case, we were happier with our original title. In case you feel the title was not a problem, we would prefer to recover the original one.

Second, we have made our best to explain the methodology in detail by means of many examples based on synthetic datasets (see Figures 1 to 4 and Table 1). This was suggested by both Reviewers and it was a great idea, which has improved substantially the manuscript.

Third, we have made an effort to reduce the part describing the datasets and the methodology without sacrificing the thoroughness of the description of the methodology, which has been also improved by adding new equations and explanations. This addressed concerns raised particularly by Reviewer #2. Thus, we hope that the methodology is completely understood in the current version of the manuscript.

We have also discussed in detail the concerns raised by Reviewer #1 about the sensitivity of the results of the diagram to the degeneracy of eigenvalues (see end of section 4.1). This has triggered the inclusion of more diagnostics in the outputs of the functions of the SailoR package, which will be available in its next version.

Since neither of the reviewers complained about the novelty of the diagram and since we have successfully addressed all points raised by the reviewers, we hope the manuscript can be accepted in the current status.

Thanks in advance. We are looking forward to your decision.

Jon Sáenz on behalf of all coauthors.

**Reply to comments by Reviewer#1**

Text by reviewer is written in red, our reply in blue.

The paper addresses a relevant and often appearing issue: comparing vector quantities. It reviews the different approaches developed so far, giving appropriate credit to those, and adds the idea of a novel graphics presentation as "sailor diagram".

This is potentially a useful tool for a vast range of applications, several examples are chosen from different fields for illustration. The deviation of method is clearly outlined and valid, reproducibility is excellent. The title is excellently chosen, abstract is concise and the term "sailor diagram" justified in the paper. Language and maths are clear. Figures are less clear. Grey squares in all figures are hard to spot (and important). Althoughit is nice that the figures relate to real world examples, for introducing the conceptit would be helpful to have figures showing clearly the benefits and limitations of the sailor diagram. Figure 2a is a very good one. The others are not easy to interpret, i.e., helping less to understand the concept. Applicability and interpretation, and general presentation would benefit from clearer examples. Given that the graphics are a central idea of the paper, following revisions are suggested, with the intention to improve understanding and uptake of the Sailor diagram for other researchers:

We sincerely appreciate the comments in this review.

1) List and number the features of the Sailor diagram clearly, eg., like i) size of ellipse depicting covariance ii) direction of ellipse indicating error main axis iii) squares indicating bias for both components iv) options for scaling as indicated in Fig. 1
2) give one (possibly synthetic) example figure illustrating clearly each feature (e.g.,datasets disagreeing on i) and agreeing on ii) and iii). For iv) note what scaling comes with which advantage / disadvantage.

We answer points 1) and 2) above in the same places of the manuscript. As we already replied in our author comment during the interactive discussion, we have created some synthetic datasets and the corresponding figures allow us to show these individual sources of error in detail, one by one, as we develop the methodology section.

We have obtained a one-year long dataset of hourly zonal and meridional wind components in the extratropical Pacific from ERA5. From this dataset, considered as Reference, we have produced different synthetic datasets with individual sources of error artificially added to them. In the first case (MOD1), we add a constant bias. Next, the error is produced (MOD2) by rotating the zonal and meridional components by 30º counterclockwise (thus also inducing a bias due to the rotation of the mean vector). An unphysical source of error is added in MOD3 by randomly resampling the dataset in order to break the original correlation of the vectors while keeping the bias and EOFs at their original values. Finally, we multiply the original data by a constant in order to change the variance of the data (MOD4), although the scaling produces a change in the bias, too. In order to reply these comments by Reviewer #1 (and additional comments by Reviewer#2), we have used these synthetic datasets in different figures in order to describe step by step the way the Sailor diagram is created. These steps are described in Figures 1, 2, 3 and 4 of the new manuscript and Table 1. This has lead to a much improved version of the manuscript, since the methodology is now better explained.

3) Explain the underlying assumptions and the limitations (i.e., what could go wrong with the interpretation). For instance, in Fig 1, the almost orthogonal major axes – are they caused by the two EOF being approx. same size and some noise deciding on whether the correct EOFS are aligned in the graphics? Are Fig. 1 (major and minor axis) thus showing a possible pitfall of interpretation of the Sailor diagram? What other limitations and possible pitfalls do exist ?

The reviewer is right in the sense that the eigenvalues of the covariance matrix are close, even though they are not exactly equal. In the current version of the manuscript, we think we address this comment by explicitly writing in the text some of these potential pitfalls. Additionally, in order to better identify the one pointed out by reviewer (degeneracy of eigenvalues), we have added two new diagnostics to the package (eccentricity of the ellipses for all datasets and congruence coefficients of the EOFs). When the eccentricity is close to zero, the user is advised not to be very confident in the evaluation of the directional results. The congruence coefficient is a fast and easy measure to identify whether the EOFs match each other, which makes a better job than relative rotation alone.

We describe these new diagnostics in detail in the new version of the manuscript (Section 3.2, pages 9 and 11). However, we must remember that the Sailor diagram is built around the concept of RMSE, that this value is always presented in the diagram and that the RMSE does not depend on the relative rotation of the eigenvectors selected by the diagonalization procedure. It can be computed directly from the series. Thus, we appreciate this comment and we discuss this in detail in the current version of the manuscript (sections 3.1 and 4), but we do not see that the results are only valid for different eigenvalues. In order to allow the user to make a proper diagnostic of rotation of the EOFs, the congruence coefficient has also been added to the set of diagnostics of the package (section 3.2).

4) Remove figures not adding information. The whole section 2 (data description) is not necessary for the understanding of the principle of the Sailor diagram and can be shortened significantly, just serving the understanding of real world examples. It is not clear for what Fig. 3a is needed – and its explanation is full of abbreviations (check"per" and "pers"). Somebody not familiar with these particular data sets cannot extract sensible information from section 4.4.

Following the same advice by both reviewers, we have significantly reduced section 2 to two thirds of the original length even though we have included a new (synthetic) dataset in order to better explain the methodology, as suggested by both reviewers. We still keep a more elaborated variable such as vertically integrated moisture transport besides wind and current because we think it is illustrative of the potential of the diagram to be used with any kind of vector quantity beyond the obvious ones (wind and current). For us, this is a very important value of the paper. We have correspondingly changed the ordering of Figures and parts of sections 3 and 4 that were affected by these inclusions or removals of datasets.

5) Figure 4 (right) needs clarification. It is impossible to relate the color codes to the 2 clusters of ellipses. Why are there exactly 2 clusters of ellipses? Furthermore, it is unclear what the centres should denote. Why are there 2 grey ellipses in the upper cluster? It is unclear what is intended to show. I cannot draw conclusions from this figure. Either clarify or remove this figure.

We have changed the colours used in the ellipses, but the problem of their visibility is actually related to the small intra-ensemble bias. We explain in detail in the current version of the manuscript that the fact that the ellipses can not be distinguished is due to the small intra-ensemble bias/change in the EOF structure of the different members of the ensemble. This is (in our opinion) a positive result and a practical example of an interesting diagnostic obtained from the sailor diagram instead of a weakness of this representation. However, as we discuss in the paper, this is one of the ways to analyse ensembles. In the second option, if all the realizations are lumped into a single ellipse (probably the best way to analyze ensembles from the theoretical point of view), this problem does not happen. The user is the one who will decide whether the ensemble must be analyzed according to the first or the other second methodology, depending on the target of the study.

6) It is commendable you provided an R package SailoR. It would be good to state clearly in section 3.7 which figure is included in the manual (instead "some of these plots")

We detail that in the current version of the manuscript and the manual of the new version of the package.

7) For better visibility, consider plotting the squares on top of the lines, to change grey to black, and to enlarge the size of the squares.

The current version of the package plots the diagrams this way. We have also changed the type of points (filled or empty) for the centered version of the diagram to improve visibility. We agree with the reviewer that visibility is increased this way. We will upload this new version to CRAN after the paper is accepted, so that we can include the DOI of the journal article in the next version of the package in the CRAN archive. The package can always be accessed at the DOI: 10.5281/zenodo.3543717 before we submit it to CRAN when the paper is accepted.

**Reply to comments by reviewer#2**

Text by reviewer is written in red, our reply in blue.

Comments on the manuscript entitled "The Sailor diagram. An extension of Taylor's diagram to two-dimensional vector data" by Sáenz et al. submitted to GMD Recommendation: Major revision

The comments in this review have been helpful to identify points of the paper where better explanations were needed. Thank you.

Summary comment The authors aim to propose a model evaluation method in terms of vector data. They constructed a "Sailor diagram" and claimed that this diagram is an extension of Taylor diagram. In my point of view, it is very farfetched to say the Sailor diagram is an extension of Taylor diagram. The Sailor diagram is not even like Taylor diagram. Two diagrams presents very different statistics. For example, Taylor diagram can illustrate correlation coefficient, standard deviation, and RMSE. However, the Sailor diagram shows the first and second EOF with the semi-major and semi-minor axes of ellipses, respectively. Each ellipse represents one model or observational data, the difference between model and observation is judged visually, which is less objective. More importantly, why are the first and second EOFs useful and what is the implication of the EOFs in terms of climate model evaluation? The EOFs between model and observation may represent different patterns. In this case, the comparison between model and observation can give wrong conclusions. These questions were not clearly interpreted (addressed) in the manuscript. Substantial revision is needed before the manuscript can be considered for publication in GMD. Detailed comments are listed below.

We have made our best in order to improve the explanations required by the reviewer. We hope that the new manuscript explains much better the methodology that we use. We were perhaps not communicating it very effectively in our previous version. However, we disagree with him/her in some points, as explained below and at large in the commentary that we submitted to the interactive discussion.

- We do not agree with the reviewer that the sailor diagram does not present the same statistics as the Taylor diagram. He explicitly mentions correlation coefficient, standard deviations and RMSE as the variables that should be present in the diagram.
  - Regarding **RMSE**, all the Sailor diagrams that we include in our paper show the exact value of the RMSE as a legend. It is a fundamental part of the diagnostic, and it is exact. The ellipses help in the interpretation, but the final element in the discussion is the RMSE, it is a numeric value, included in the diagram, and this result is numerically exact. Thus, we can not accept the words by reviewer above "*the difference between model and observation is judged visually, which is less objective*". No, the comparison in the sailor diagram is numerical and exact. As shown in our paper, Tables 1 and 2 show that our methodology provides additional diagnostics which are even more relevant that the RMSE alone (the relative rotation of the ellipses, for instance).
  - There is not a universally accepted definition of **correlation in two dimensions**, as we discuss in the paper. The most accepted one involves the sum of two different correlations (canonical correlations). Thus, we have

preferred to keep different pieces of information (like the eigenvalues and eigenvectors of the covariance matrix) in our diagram, since they provide additional and powerful diagnostics to it. The addition of the canonical correlations to the diagram would clutter it too much (as shown by our internal tests) and we prefer not to do that. As shown by Tables 1 and 2 in the paper, the correlation based in the canonical correlations can be retrieved from the package for all the models in tabular form. That this table is less informative than the Sailor diagram is clearly seen in current Table 1, in the column corresponding to $R^2$ from the different models. Since the correlation coefficient based on canonical correlations is by definition invariant to rotations, MOD3 yields a perfect value for $R^2$. However, a model such as MOD3, with rotated vectors, must obviously be assigned a lower skill than an alternative one which does not rotate them. This result is explained in a new paragraph in section 3.2 and 3.3 (lines 343-349).

○ The diagram effectively and graphically represents the **standard deviations** of the model and observations, by means of the corresponding ellipses. They represent the standard deviations of observations and models as the length of the semi-major and semi-minor axes. They are shown in the diagram and can be visually compared. The decomposition of the datasets in terms of the EOFs of the two-dimensional covariance matrix of the zonal and meridional components is fundamental to achieve this. The basic idea behind the Taylor diagram (and the Sailor diagram) is that the visual presentation must allow a fast intercomparison of models. The EOFs play a central role in allowing a fast evaluation of the directional properties of the data. We have improved the explanation of the diagram (the link between the equations and a step-by-step description of the way it is built) to make it clear in the current version of the paper.

● We do not follow the original design of the Taylor diagram which addressed scalar variables, and we already explicitly recognized it in the text of the original version of the manuscript. The reason is that we thought (and still think) that presenting additional diagnostics related to the directional properties of vector variables (principal axes and their corresponding standard deviations) was more informative than keeping the structure of a diagram designed for scalars. We still think it is a better idea to keep this information for a vector quantity in a new version of a diagram designed from scratch addressing these specific needs of vector time series or fields.

We have made our best to make these points clear in the current version of the manuscript.

1. The title of the manuscript is misleading and should be changed because the Sailor diagram is totally different from the Taylor diagram. Two diagrams present very different statistics and have different implications. For example, Taylor diagram presents correlation coefficient, standard deviation, and RMSE. However, the Sailor diagram was constructed based on the EOF of vector data, which does not explicitly include correlation coefficient and standard deviation. Two diagrams do not look like each other, either.

As we already explained in our author comments during the interactive discussion, we do not agree with reviewer # 2 here. The Sailor diagram provides two-dimensional standard deviations (axes of the ellipses), RMSE in a legend and correlation coefficients based on CCA can also be obtained from the implementation we provide in a tabular data. The

change in design is due to the fact that it was our intention to provide an additional feature, which is the ability of the diagram to make a full analysis of the structure of the errors in direction and standard deviations of the vectors as well. We find our diagram very efficient in achieving this, without resorting to any normalization of the vectors. Thus, we do not agree with the reviewer here. Besides that, we already provide a package in CRAN which is called SailoR, so that we find no reason to change the name of the diagram. We have changed the title of the paper removing any mention to Taylor's diagram. However, we would prefer to keep the original one, and we will recover it if editor agrees. We will in any case keep the name of the package and the diagram as Sailor in CRAN.

2. Section 2 introduced five different vector datasets using 3 pages. It's not necessary to use so much dataset and can be reduced since they are all vector data. Only one or two of them should be enough to interpret the diagram. In contrast, methodology (section 3) is the key part of the manuscript which should clearly interpret and explain the method. However, the methodology was not well interpreted and hard to follow. I suggest that the authors interpret the methodology using an example data. Eu, Pu, and EOFs can be illustrated by using the example data to help readers to understand the method. The method and its application should be clearly interpreted in terms of model evaluation. In addition, section 3.1 and 3.2 generally present the same equations and can be merged.

We have substantially shortened this section even though we still keep a minimum number of datasets which illustrate the methodology (the new synthetic dataset requested by the reviewers) and five different datasets to illustrate the use of the diagram in different setups (three for time-series analysis), one for spatial fields and a last one for ensembles. We find that removing additional datasets would severely harm the information conveyed by the paper. In particular, we think it is interesting to note that the use of an additional variable such as vertically integrated moisture transports allows the reader to consider that the methodology can be applied to other vector variables beyond the obvious ones (current and wind).

We have also made our best to shorten former section 3.2 without sacrificing the explanation of the next steps of the algorithm (the use of the basis formed by the EOFs of U in the definition of the V anomalies). We have removed it (except one paragraph), so that we are still able to analyze the relative rotations of the datasets. We have added new figures (Figure 1 to Figure 4) to illustrate the concept of principal axes, rotation and bias in current sections 2 and 3 of the manuscript. We have also introduced the methodology behind the manuscript step by step in these Figures 1 to 4 and Table 1 (as suggested also by Reviewer # 1), so that we hope everything is clear now.

3. Line 49-51 and 78-80: This is not true. The Taylor diagram can be extended to two (even more) dimensional vector data evaluation directly by using a set of statistical quantities defined by Xu et al. (2016). This paper was also cited by the authors.

We have rewritten these sentences.

4. Line 99-101, 109-112: To my knowledge, Xu et al. (2016) did normalize various statistics but no approximation was applied. The author argued that the merit of the Sailor diagram is that no approximation is needed. However, Sailor diagram illustrate the first two EOFs. Each EOF can only explain part of the variability of the original data.

We have changed the sentence to avoid any reference to Xu et al. However, we disagree with the reviewer when he/she mentions that our diagram doesn't explain the full variability of the data. With a long enough sample (even very short series of real wind would be enough), the covariance matrix that we use to define the EOFs will be of full rank, since it is a two-dimensional covariance matrix (see equation 5 in our paper). Thus, the leading two EOFs will always explain 100% of the total variance for any sensible and realistic geophysical flow (unless it is stationary and laminar). We have improved the explanation of the algorithm in the current version of the manuscript, so that we hope it is now clear that this is the case.

5. Line 220-223: It is confusing that the authors use "U" to represent observation and "V" to represent model because U and V were usually used as the zonal and meridional component of wind. I suggest the authors replace "U" with "O" and "V" with "M" or other appropriate notation to avoid confusion.

As we responded during the interactive comments, we feel the current notation is not a problem for the understanding of the paper (Reviewer#1 agrees on that). However, we have improved the explanation at the beginning of the methodology part when referring to the structure of the U and V matrices, so that it is now even better explained than in the initial version of the manuscript.

6. Line234: How are the principal components of the data standardized?

They are not standardized at all. Maybe the reviewer was mislead by our use of two matrices, one which holds the variance-carrying principal components and a second one with standardized principal components. However, this scaling is only used for principal components in some equations when they are also multiplied by the corresponding standard deviations (see equations 1, 3, 7, 10 and 23 as good examples), so that no variance is lost in the process. We have added a sentence to the manuscript (line 209) to make this clear. Additionally, equation 10 describes the way the ellipses are derived, so that it is completely clear that no standardization is applied in the current version of the manuscript.

7. Line 360-361: What is the implication of the relative rotation between EOFs from observations and simulations? Why is it important to model evaluation?

As we explained during the interactive comments, it is important that both model data and observations show similar major/minor directions (axes) and similar standard deviations in both axes. We have improved the explanation about this in the current version of the manuscript. See pages 10, Table 1 and figures 2 and 4. We hope the reason for this is clear now.

8. Line 361-363: Why the variance explained by each EOF is important in terms of model evaluation? What if the EOFs between model and observation represent different patterns?

As we pointed out during our replies in the Interactive comment, we are using a two-dimensional covariance matrix, so that the concept of pattern for the EOFs which is common in the decomposition of the time and spatial patterns of variability in climatology does not apply here. The variances corresponding to each EOF represent the length of the principal axes of the ellipses used to represent the data corresponding to model and

observations. We have added equation 10 to make that explicit. Thus, if the model data represents properly the observations, the lowest RMSE is expected. By means of equations 18, 19, 21 and 27, a perfect model implies that the bias is zero and the ellipses are the same. EOFs do not represent in our case a spatial pattern, but a pair of orthogonal directions (an orthonormal basis) in the horizontal plane formed by zonal and meridional components as shown by equation 6 and explained now thoroughly in lines 221 to 225 of the manuscript. We hope that with the current version of the manuscript and the additional explanations and figures we provide in the methodology section everything is clear now.

**Compare Results**

Old File:

**saenz-GMD-2019.pdf**

**24 pages (346 KB)**

12/10/2019 19:35:04

versus

New File:

**saenz-GMD-2019r01.pdf**

**27 pages (1,73 MB)**

01/02/2020 9:39:15

**Total Changes**

**681**

**Content**

286    Replacements

153    Insertions

116    Deletions

**Styling and Annotations**

22    Styling

104    Annotations

Go to First Change (page 1)

[revised manuscript text omitted]

---

## Author Response (AR2)

Dear Editor.

Please, find enclosed the revised version (R02) of original manuscript "**The Sailor diagram. An extension of Taylor's diagram to two-dimensional vector data**", currently entitled "**The Sailor diagram. A new diagram for the verification of two-dimensional vector data from multiple models**" by J. Sáenz, S. Carreno-Madinabeitia, G. Esnaola, S. J. González-Rojí, G. Ibarra-Berastegi and A. Ulazia, https://doi.org/10.5194/gmd-2019-289

Together with this cover letter, we are submitting the revised manuscript. In it, we have added to all sections of the paper (Abstract, Introduction, Methodology, Results and Conclusions) paragraphs or sentences making it very clear that no truncation is happening in our paper. Thus, we hope that we are successfully giving a proper answer to your last requirement. We have made explicit in a new structure for Table 1 of the new version of the paper that the variance explained by EOFs and the original wind series is exactly the same. Some values have changed in this Table due to the fact that MOD3 is built by means of a random resample of the original dataset. However, the exact data produced by the same permutation of the sample used in the paper are already available in the future version package, so that users will get exactly the same results as in the paper when we release it (hopefully, after the paper is accepted). The new version of the package can be downloaded from the URL:
https://zenodo.org/record/3654994/files/SailoR_1.1.tar.gz?download=1

We have also realized that in Figure 4 (left) of the previous version of the revised manuscript one of the ellipses touched the legend of the plot. Thus, we have slightly changed its axes so that this does not happen now.

Together with the revised paper, we are also including a PDF document with the changes introduced in the new version of the manuscript highlighted.

As we already mentioned in the cover letter that we submitted with the previous reviewed version of the paper, we would like to draw your attention to the fact that, because of the requirement by Reviewer #2, we changed the title of the paper to "**The Sailor diagram. A new diagram for the verification of two-dimensional vector data from multiple models**". In any case, we were happier with our original title. In case you feel the title was not a problem, we would prefer to recover the original one when the paper is accepted. But if it is difficult at this point, we would just keep the new title.

Thanks in advance. We are looking forward to your decision.

Jon Sáenz on behalf of all coauthors.

Response to editor in Review#2 of manuscript: "**The Sailor diagram. A new diagram for the verification of two-dimensional vector data from multiple models**" by J. Sáenz, S. Carreno-Madinabeitia, G. Esnaola, S. J. González-Rojí, G. Ibarra-Berastegi and A. Ulazia, https://doi.org/10.5194/gmd-2019-289

Dear Dr. Marti.

Our answer is written in blue. The original text by the editor is written in red.

Revised Submission
Topical Editor Decision: Reconsider after major revisions (18 Feb 2020) by Olivier Marti
Comments to the Author:
Dear authors,

Both reviewers have very different opinions about your paper. I send the paper for further review(s), but I'm asking you a revision of the manuscript first.

The mathematics of the paper is based on an EOF decomposition of time series of vectors. The dimensionality of the problem (2xN) is such that U,V,... are fully described by only 2 EOFs of size N. These two EOFS explain the full variability, with no loss of information. The method used for spatial data keeps this property of a signal fully described by the two EOFs. This should be a mathematical evidence. But in the present wording of the paper, this is a point that could be easily missed by a reader.

We have significantly expanded the text in many parts of the paper to effectively correct this weakness of the paper. We have added a sentence to the abstract to make that clear from the beginning. We have written a paragraph in the introduction covering the most usual way the EOFs are applied (lines 113-123). In this paragraph we explicitly mention the way reduction of variance is commonly achieved in climatology by reducing the number of EOFs below the rank of the covariance matrix. Later, in page 8 (subsection 3.1), we explicitly write the covariance matrix of the data by following the notation that we use in the paper. We show that the covariance matrix is of full rank and it is a 2x2 matrix. We explicitly mention (lines 229-232) that there is no truncation because the same number of EOFs (two) as the rank of the covariance matrix are kept for the analysis. In subsection 3.2 (page 15, Table 1), we have re-structured the columns of the table so that we compare the variance of the original zonal and meridional components with the one computed from EOFs. It is show the variance is the same, and this is mentioned in lines 357-358 (page 14). We insist again on it in page 23 (lines 532-533) when presenting the results corresponding to ensembles.

As this point is key for a good understanding of the Sailor diagram, it should be made very clear with no ambiguity of any kind for any reader.

We sincerely think that there is no ambiguity now in its current version. However, we agree that it will make its interpretation easier for future readers.

I send the paper for a major revision to allow you to fully clarify this point, to be sure that no future reader will misunderstand this lossless decomposition.

Best regards, and thank you for submitting in GMD.

Olivier Marti
GMD Topical Editor

We hope that the current version of the paper correctly addresses your concerns.

Jon Sáenz on behalf of all coauthors.

**Compare Results**

Old File:

**saenz-GMD-2019r01.pdf**

**27 pages (1,73 MB)**
01/02/2020 9:39:15

versus

New File:

**saenz-GMD-2019r02.pdf**

**29 pages (1,75 MB)**
25/02/2020 16:56:41

**Total Changes**

**462**

**Content**

99 Replacements

181 Insertions

146 Deletions

**Styling and Annotations**

11 Styling

25 Annotations

Go to First Change (page 1)

[revised manuscript text omitted]

---

## Author Response (AR3)

**Submitted on 14 May 2020, Anonymous Referee #3**
**Review of "The Sailor diagram. A new diagram for the verification of two-dimensional vector data from multiple models", by Sáenz et al.**

This paper presents a graphical way of comparing two dimensional fields (like wind speed), like Taylor diagrams, designed to compare model simulations.
The methodology is well described and the applications are clear.

Thank you for these comments. Our answers below will be typed using blue font.

I have a few concerns on the methodology:
The way the results are shown implies using colors, which prevents its use by color blind researchers. For example, the lines in Figs. 6b, 7b, 8 are very hard to distinguish.

We have adapted the package so that the current version has two additional arguments giving it the ability to change the linetype and the characters used to mark the location of the means. This way, the Sailor diagrams can also be prepared using black and white. We have changed figures 7a and 7b to show these new capabilities of the software. We have also added a new sentence explaining this to the end of section 4.4. This new version of the software will be uploaded to the CRAN repository when the paper is accepted, so that it can be referenced by DOI. By now, the new version of the software can be retrieved from the following DOI: **https://doi.org/10.5281/zenodo.3543717**

Taylor diagrams can easily show results for tens of models. Sailor diagrams for more than two models might be awkward to decipher.

We have made our best to improve the interpretability of the Sailor diagrams. It is true that they are more complex than the original Taylor diagrams. However, the reviewer must consider that they also convey relevant information such as the errors in directionality or the fraction of variance associated to the principal axes of the covariance matrix which can not be diagnosed in the original Taylor diagrams, but which are important for vector variables. In any case, we stress that the design of the diagram offers two additional degrees of freedom which allow the user to prepare efficient diagrams:
- **Scale parameter**: Please, compare figures 5a versus 5b or new figures 9a versus 9b. The use of a scaling parameter allows the user to assess the relative biases besides the rotation/fraction of variance parameters in the individual ellipses.
- **Centered versus uncentered diagrams**: The impact of using this centered/uncentered approach is clearly seen when comparing Figures 4a versus 4b and 8a versus 8b in the new version of the paper. When a centered version of the diagram (4b, 8b) is prepared, the number of lines corresponding to the reference dataset is severely diminished. Thus, centered diagrams allow an easier interpretation of errors which appear due to the rotation of semi-major or semi-minor axes or the different fractions of variance associated to them.

We have added some paragraphs to make this clear in the current version of the manuscript, section 4.5, pages 23 and 24 and Conclusions, page 25. In order to take the Sailor diagram to the extremes, we have included all the realizations available from the CMIP5 experiment for the MIROC and IPSL models, and we have added a new model (HadGEM2-ES). We have restructured Figure 8 and prepared a new Figure 9 with 15 realizations to discuss the use of these parameters in detail.

Besides the previous consideration, please note that if 2D vectorial magnitudes are to be compared using Taylor diagramas you need to duplicate the number of diagrams,with one set of taylor diagrams for one dimension and a second set for the another dimension. In our previous works we have extensively adopted that solution but precisely to overcome this difficulty we have developed the Sailor diagram. The use of doubled Taylor diagrams does not provide a combined estimation of error.

Sailor diagrams do not directly provide the values of the diagnostics (unlike Taylor diagrams). This caveat could be discussed in the paper, even briefly.

We can not completely agree with the reviewer here. It is true that the angle with the abscissa is not related to the correlation coefficient, or that the RMSE error is more complex than in the case of the Taylor diagram. On the other hand, the two components of the bias are clearly shown. The errors in the directionality are also graphically shown by the orientation of the semimajor and semiminor axes of the ellipses, and these parameters are easily analyzed visually. The fraction of variances corresponding to the principal axes to each dataset are also graphically shown. Besides that, as we suggest in the paper, a legend with the exact value of the RMSE error is also provided in the diagram.

We think that, besides the diagram (which we find useful), the structure of the diagnostics shown in the paper can also be expressed in a tabular form, and the full set of diagnostics is provided by our implementation. Some of them are shown in Tables 1 and 2 of the paper. These diagnostics are exact, as discussed by the methodology discussed in the paper. The diagrams are great to achieve a visual evaluation of the bias and the errors in directionality. However, if a sensible use of the scale factor and a centered/uncentered diagram (see the reply to previous comment) is not enough to make the diagram interpretable because the number of models to be evaluated is too large, a first screening by using a selection of columns in tabular form can be followed by a diagram for a subset of the better models in the table is a different option. We have added a new paragraph in the conclusions mentioning this.

My second issue is the apparent complication of using 2d EOFs. Why not compute the correlation between complex numbers, as 2d fields can be represented as complex numbers? Hence, one retrieves the phase and amplitude of the (complex) correlation. And then produce something like Taylor diagrams. I am not saying that Sailor diagrams are a bad idea or wrong… but a relatively simpler formulation with complex numbers would allow using the existing machinery of multivariate spectral analysis, and make a use of Taylor diagrams. I won't ask the authors to do this (or even perform a comparison, although this is conceptually easy), but at least justify their option.

We appreciate this suggestion by the reviewer, since this was our initial idea when we started thinking in an alternative to the Taylor diagram. However, we decided to progress through a different route. When reviewing literature during the early stages of this project, we realized that there were different definitions of the complex correlation coefficient, see eg. Hanson et al., (1992) and Schreier (2008). One of them is closely related to the definition of $R^2$ by other references that we already used and discussed in previous version of the paper in the introduction section (see, for instance, Jupp and Mardia (1980)).

Hanson, B., Klink, K., Matsuura, K., Robeson, S. M., and Willmott, C. J.: Vector correlation: Review, Exposition and Geographic Application, Annals of the Association of American Geographers, 82, 103–116, 1992.

Schreier, P. J.: A Unifying Discussion of Correlation Analysis for Complex Random Vectors„ IEEE Transactions on Signal Processing, 56, 1327–1336, https://doi.org/10.1109/TSP.2007.909054, 2008.

Besides the lack of a universally accepted version of the correlation coefficient for two-dimensional variables, we also decided to define a new structure for the Sailor diagram because we thought that it would be more informative than just transposing the structure of the Taylor diagram to vector quantities. We quote from our previous version of the manuscript (lines 104-112 of previous version of the paper), with bold face in some points:

> Thus, **we have decided to follow a new approximation** which does not lead to the common Taylor diagram used for scalars, but **gives more information about the structure of the two-dimensional errors** between vector quantities involved in the verification of a vector quantity derived from a model with its observational counterpart (reference dataset). This is the rationale which leads us to base our definition in a full use of the two-dimensional structure of the mean squared error (MSE) between both vectorial datasets. This does not allow us to reduce our diagram to the well-known Taylor diagram used for scalars, as the one produced by Xu et al. (2016). However, we hope that our diagram will be considered a valuable contribution to the set of techniques used for the evaluation of models, as **it visually explores other properties of the error** between the vector datasets, such as the **relative rotation** of the major axes of variability and the **underestimation (or overestimation) along each principal axis** of the covariance matrix.

Thus, we make clear in our paper that we give more value to finding an alternative visualization of the error which makes it easier to analyze problems in directionality or explained variances rather than just to replicate the structure of the original Taylor diagram, which was designed with scalars in mind. For us, it was important to use the EOF-based decomposition of the two dimensional covariance matrix in order to improve the interpretability of results because of the following reasons:

- The use of EOFs leads to a better assessment of the main directions represented by models (this allows us to evaluate the errors in directionality).
- Assessing the length of the main axes of the ellipses (and their eccentricities) allows us to compare the fractions of variances from each model to the ones from observations.
- EOFs lead naturally to one of the most accepted definitions of $R^2$, the one based by the sum of the squared canonical correlations (see Jupp and Mardia (1980) or Crossby et al. (1993) to name a few already cited in the paper).

We have rewritten this paragraph in the new version in the hope to make clear the message that we have changed the structure of the original Taylor diagram because we want to diagnose errors in directionality, which are very important for vector quantities. We want to stress now that these diagnostics would be lost if we didn't use as a starting point of our analysis the 2x2 mean squared error matrix in Eq. 21. We have added a short paragraph stating this in the new version of the paper. The major role played by the use of EOFs versus $R^2$ (akin to complex correlation for one of the possible definitions) is made clear by the example MOD2 in the synthetic datasets. As shown by Table 1, $R^2$ for MOD2

is 2 (perfect squared correlation) even though the dataset is rotated 30º as properly identified by our rotation parameters. The reason is that $R^2$ properly evaluates the linear dependence between two datasets, and rotation implies a linear dependence. However, it does not identifies the value of the rotation, which is important when comparing vector quantities. This is properly explained at the end of section 3.3.

Minor comments
What is the use of Fig. 1a?

During the early revisions, one of the reviewers (see comment 2 by Rev#1) suggested us to explain better the algorithm by using synthetic datasets. We added this figure to improve the readability of the manuscript. This is also related to the comments by Reviewer#2. It was designed to properly explain the readers the central role played by EOFs. For many readers it might be clear, but for others it, apparently, is not. In particular, some readers might find hard to understand the reason that our use of EOFs does not imply any truncation in the datasets. We found this suggestion by Reviewer#1 excellent and we would prefer to keep this figure. We have added a sentence to explain the reason behind our decision to include of this figure. If in this new version the editor suggests us so, we could remove it, but we think it helps in the interpretation of the diagram setup.

Should I deduce from Fig. 8 that the two models have the right wind direction?

If you mean by "right wind direction" the direction corresponding to EOFs, the answer is yes, but we have realized this must be better explained. We have added some sentences to the interpretation of this Figure in Section 4.5, page 22 of the version with changes marked. We thank this comment, since it allows us to arrive to a better explanation of our results.

The diagnostics on the MIROC and IPSL simulations reveal that one is positively biased, the other is negatively biased, but the difference between the two models is <0.03 m/s. Is this meaningful? I would have prefered that the authors chose examples where the differences are really important, which would give an incentive for bias correction.

We only wanted to show that ensembles could also be processed by using the Sailor diagram. We selected for this case global climate models and we wanted to verify climate models, which was the original idea behind the Taylor diagram. We have added a third model (HadGEM2-ES) to Figures 8 and 9 and have increased all the realizations available from all the models. However, since we are dealing with the long-term (27 years) averaged surface wind over the Southern Hemisphere, the errors can not be great. With the addition of the new model, the sign of the bias changes and this allows us to better explain the characteristics of these figures. We find that the corresponding figures show important characteristics of the ensemble, such as the inter-model and intra-model variabilities.

Selecting other examples from operational Ensemble Prediction Systems is a different option, but we find it wouldn't add any important conclusion to the paper.

[revised manuscript text omitted]